# HINDSIGHT FORESIGHT RELABELING FOR META-REINFORCEMENT LEARNING

**Michael Wan, Jian Peng & Tanmay Gangwani**
University of Illinois at Urbana-Champaign
{mw3,jianpeng,gangwan2}@illinois.edu

## ABSTRACT

Meta-reinforcement learning (meta-RL) algorithms allow for agents to learn new behaviors from small amounts of experience, mitigating the sample inefficiency problem in RL. However, while meta-RL agents can adapt quickly to new tasks at test time after experiencing only a few trajectories, the meta-training process is still sample-inefficient. Prior works have found that in the multi-task RL setting, relabeling past transitions and thus sharing experience among tasks can improve sample efficiency and asymptotic performance. We apply this idea to the meta-RL setting and devise a new relabeling method called Hindsight Foresight Relabeling (HFR). We construct a relabeling distribution using the combination of *hindsight*, which is used to relabel trajectories using reward functions from the training task distribution, and *foresight*, which takes the relabeled trajectories and computes the utility of each trajectory for each task. HFR is easy to implement and readily compatible with existing meta-RL algorithms. We find that HFR improves performance when compared to other relabeling methods on a variety of meta-RL tasks.

## 1 INTRODUCTION

Deep Reinforcement Learning (RL) has achieved success on a wide variety of tasks, ranging from computer games to robotics. However, RL agents are typically trained on a single task and are extremely sample-inefficient, often requiring millions of samples to learn a good policy for just that one task. Ideally, RL agents should be able to utilize their prior knowledge and adapt to tasks quickly, just as humans do. Meta-learning, or learning to learn, has achieved promising results in this regard, allowing agents to exploit the shared structure between tasks in order to adapt to new tasks quickly during meta-test time.

Although meta-learned policies can adapt quickly during meta-test time, training these meta-learned policies could still require a large amount of data. Several popular meta-RL methods (Duan et al., 2016; Finn et al., 2017; Mishra et al., 2017; Rothfuss et al., 2018; Wang et al., 2016) utilize on-policy data during meta-training to better align with the setup at meta-test time, where the agent must generate on-policy data for an unseen task and use it for adapting to the task. Recent works (Fakoor et al., 2019; Rakelly et al., 2019) have sought to incorporate off-policy RL (Fujimoto et al., 2018; Haarnoja et al., 2018) into meta-RL to improve sample efficiency.

The combination of off-policy RL and relabeling, in which experience is shared across tasks, has been utilized in the multi-task RL setting, in which an agent learns to achieve multiple different yet related tasks, for both goal-reaching tasks (Andrychowicz et al., 2017) and more general multi-task settings (Eysenbach et al., 2020). Experience collected for one task may be completely useless for training a policy to learn that task, but could be extremely informative in training a policy to learn a different task. For example, an agent trying to shoot a hockey puck into a net might miss to the right. This experience could easily be used to train an agent to shoot a puck into a net positioned further to the right (Andrychowicz et al., 2017).

Both meta-RL and multi-task RL involve training on a distribution of tasks, so it follows that we can also combine relabeling techniques with meta-RL algorithms in order to boost both sample efficiency and asymptotic performance. In meta-RL, an agent learns to explore sufficiently to identify

---

Code: https://www.github.com/michaelwan11/hfr

the task it is supposed to be solving, and then uses that knowledge to achieve high task returns. The agent collects exploratory pre-adaptation data, then undergoes some adaptation process using that pre-adaptation data. Finally, after adaptation, the agent attempts to solve the task. Meta-RL algorithms typically have a meta-training phase followed by a meta-test phase. The goal during meta-training is to train the meta-parameters such that they could be quickly adapted to solve any task from the meta-train task distribution, given a small amount of data from that task. At meta-test time, given a new unseen task, the goal is to rapidly adapt the *learned* meta-parameters for this task, using a small amount of task-specific data. The focus in this paper is to improve the sample efficiency of the meta-training phase via data sharing.

Using concepts from maximum entropy RL (MaxEnt RL), we introduce a relabeling scheme for the meta-RL setting. Prior relabeling methods for multi-task RL have used the total reward of the trajectory under different tasks to guide the relabeling (Eysenbach et al., 2020; Li et al., 2020). Direct application of this type of relabeling to the meta-RL setting is potentially sub-optimal since the multi-task RL and meta-RL objectives are distinct (learning to perform many tasks vs. learning to learn a new task). Towards developing an approach more suited to meta-RL, we define the notion of the *utility* of a trajectory under the different tasks, where the utility captures the usefulness of the trajectory for efficient adaptation under those tasks. We call our method Hindsight Foresight Relabeling (HFR) – we use *hindsight* in replaying the experience using reward functions from different tasks, and we use *foresight* in computing the utility of trajectories under different tasks and constructing a relabeling distribution over tasks using these utilities. We demonstrate the efficacy of our method on a variety of robotic manipulation and locomotion tasks. Notably, we show that our method, as the first meta-RL relabeling technique (applied during meta-training) that we are aware of, leads to improved performance compared to prior relabeling schemes designed for multi-task RL.

## 2 RELATED WORK

Meta-learning, or learning to learn (Baxter, 1998; Naik et al., 1992; Schmidhuber, 1987; Thrun & Pratt, 1998), has been a topic of interest since the 1980s. Various approaches have been developed in recent years. Prior works have attempted to represent the RL process using a recurrent neural network (RNN) (Duan et al., 2016; Miconi et al., 2018; Wang et al., 2016) - the hidden state is maintained across episode boundaries and informs the policy as to what task it is currently solving. Similarly, Mishra et al. (2017) also maintain the internal state across episode boundaries while incorporating temporal convolution and attention into a recursive architecture. Gradient-based meta-learning methods have also been explored (Finn et al., 2017; Nichol & Schulman, 2018; Xu et al., 2018; Zheng et al., 2020). MAML (Finn et al., 2017) seek to learn a good policy initialization so that only a few gradient steps are needed to achieve good performance on unseen meta-test tasks. Stadie et al. (2018) build on this gradient-based approach but explicitly consider the effect of the original sampling distribution on final performance. Similar to these works, our relabeling method also considers the impact of pre-adaptation data on the post-adaptation performance. Another body of work focuses on designing strategies for structured exploration in meta-RL such that task-relevant information could be efficiently recovered (Liu et al., 2020; Rakelly et al., 2019; Zintgraf et al., 2019). Rakelly et al. (2019) devise an off-policy meta-RL method called PEARL that trains an encoder to generate a latent context vector on which the meta-RL agent is conditioned. Although we use PEARL as our base algorithm in this work, our relabeling scheme is general enough to be integrated into any off-policy meta-RL algorithm.

**Experience Relabeling in Meta-RL.** Recent work has studied the scope of sharing experience among tasks in the meta-RL paradigm. Mendonca et al. (2020) propose to tackle meta-RL via a model identification process, where context-dependent neural networks parameterize the transition dynamics and the rewards function. Their method performs experience relabeling only at the meta-test time, with the purpose of consistent adaptation to the out-of-distribution tasks. Crucially, there is no relabeling or sharing of data amongst tasks during the meta-train time. In contrast, the goal of the relabeling in HFR is to improve the sample efficiency of the meta-training phase. Dorfman et al. (2020) study the offline meta-RL problem. They propose reward relabeling as a mechanism to mitigate the "MDP ambiguity" issue, which the authors note is specific to the offline meta-RL setting. Their relabeling is based on random task selection. HFR, on the other hand, operates in the online meta-RL setting and provides a principled approach to compute a relabeling distribution that suggests

tasks for relabeling. We compare with the random relabeling method used in Dorfman et al. (2020) in our experiments. A more detailed comparison to these two prior works is included in Appendix A.11.

In the context of multi-task RL (Caruana, 1997; Kaelbling, 1993; Schaul et al., 2015), recent methods have proposed relabeling to improve the sample-efficiency (Andrychowicz et al., 2017; Eysenbach et al., 2020; Li et al., 2020). HER (Andrychowicz et al., 2017) relabels transitions using goals that the agent actually achieves. Doing so allows for learning even with a sparse binary reward signal. However, HER is only applicable to goal-reaching tasks and cannot be incorporated into meta-RL algorithms because the meta-RL agent is trained on a batch of tasks sampled from a fixed task distribution. Similar to our work, Eysenbach et al. (2020) use MaxEnt RL to construct an optimal relabeling distribution for multi-task RL and apply this to both goal-reaching tasks and tasks with arbitrary reward functions.

## 3 BACKGROUND

### 3.1 REINFORCEMENT LEARNING

In reinforcement learning (RL), the environment is modeled as a Markov Decision Process (MDP) $\mathcal{M} = (\mathcal{S}, \mathcal{A}, r, \mathsf{p}, \gamma, p_1)$, where $\mathcal{S}$ is the state-space, $\mathcal{A}$ is the action-space, $r$ is the reward function, $\mathsf{p}$ is the transition dynamics, $\gamma \in [0, 1)$ is the discount factor, and $p_1$ is the initial state distribution. At timestep $t$, the agent $\pi_\theta$, parameterized by parameters $\theta$, observes the state $s_t \in \mathcal{S}$, takes an action $a_t \sim \pi_\theta(a_t | s_t)$, and observes the next state $s_{t+1} \sim \mathsf{p}(s_{t+1} | s_t, a_t)$ and the reward $r(s_t, a_t)$. The goal is to maximize the expected cumulative discounted rewards: $\max_\theta \mathbb{E}_{s_t, a_t \sim \pi_\theta} \left[ \sum_{t=1}^\infty \gamma^{t-1} r(s_t, a_t) \right]$.

### 3.2 META-REINFORCEMENT LEARNING

In the general meta-reinforcement learning (meta-RL) setting, there is a family of tasks that is characterized by a distribution $p(\psi)$, where each task $\psi$ is represented by an MDP $\mathcal{M}_\psi = (\mathcal{S}, \mathcal{A}, r_\psi, \mathsf{p}_\psi, \gamma, p_1)$. The tasks share the components $(\mathcal{S}, \mathcal{A}, \gamma, p_1)$, but can differ in the reward function $r_\psi$ (e.g. navigating to different goal locations) and/or the transition dynamics $\mathsf{p}_\psi$ (e.g. locomotion on different terrains). In this work, we consider the setting where the *tasks share the same transition dynamics (*i.e., $\mathsf{p}_\psi = \mathsf{p}$*), but differ in the reward function*. The goal in meta-learning is to learn a set of meta-parameters such that given a new task from $p(\psi)$ and small amount of data for the new task, the meta-parameters can be efficiently adapted to solve the new task. In the context of meta-RL, given new task $\psi$, the agent collects some initial trajectories $\{\tau_{\text{pre}}\}$, each being a sequence $\{s_1, a_1, s_2, a_2, \dots\}$, and then undergoes some adaptation procedure $f_\phi(\pi_\theta, \tau_{\text{pre}}, r_\psi)$ (*e.g.*, a gradient update (Finn et al., 2017) or a forward pass through an RNN (Duan et al., 2016)). The adaptation procedure returns a new policy $\pi'$. Using this post-adaptation policy, the agent should seek to maximize the cumulative discounted rewards it achieves. Overall, the meta-RL objective is:

$$\max_{\theta, \phi} \mathbb{E}_{\psi \sim p(\psi), (s_t, a_t) \sim \pi'(\theta, \phi)} \left[ \sum_{t=1}^\infty \gamma^{t-1} r_\psi(s_t, a_t) \right]; \quad \pi'(\theta, \phi) = f_\phi(\pi_\theta, \tau_{\text{pre}}, r_\psi) \quad (1)$$

where $\theta, \phi$ are the meta-parameters that are learned in the meta-training phase. A meta-RL agent must learn a good adaptation procedure $f_\phi$ that is proficient in extracting salient information about the task at hand, using few pre-adaptation trajectories $\tau_{\text{pre}}$. At the same time, it should learn the policy meta-parameters $\theta$ such that it can achieve high returns after the adaptation process, *i.e.*, while following the policy $\pi' = f_\phi(\pi_\theta, \tau_{\text{pre}}, r_\psi)$.

### 3.3 PEARL

In this work, we use PEARL (Rakelly et al., 2019) as our base meta-RL algorithm since it uses off-policy RL and provides structured exploration via posterior sampling. PEARL is built on top of Soft Actor-Critic (Haarnoja et al., 2018) and trains an encoder network $q_\phi(z|c)$ that takes in the "context" $c$, which consists of a batch of $(s_t, a_t, r_t, s_{t+1})$ transitions, and produces the latent embedding $z$. The intent is to learn the encoder such that embedding $z$ encodes some salient information about the task. The adaptation step $f_\phi$ in PEARL corresponds to generating this latent $z$ and then conditioning the policy and the value function networks on it. The policy $\pi_\theta(a|s, z)$ is trained using loss $L_{\text{actor}}$:

$$L_{\text{actor}} = \mathbb{E}_{s \sim B, a \sim \pi_\theta, z \sim q_\phi(z|c)} \left[ D_{\text{KL}} \left( \pi_\theta(a|s, z) \| \frac{\exp(Q_\theta(s, a, z))}{Z_\theta(s)} \right) \right] \quad (2)$$

where $B$ is the replay buffer. The critic $Q_\theta(s, a, z)$ and the encoder $q_\phi(z|c)$ are trained with temporal difference learning:

$$L_{\text{critic}} = \mathbb{E}_{(s,a,r,s') \sim B, z \sim q_\phi(z|c)} \left[ \left( Q_\theta(s, a, z) - \left( r + \bar{V}(s', \bar{z}) \right) \right)^2 \right] \quad (3)$$

where $\bar{V}$ is the target state value and $\bar{z}$ denotes that the gradient does not flow back through the latent.

## 4    HINDSIGHT FORESIGHT RELABELING

The objective in this section is to derive a formalism for data-sharing amongst the tasks during the meta-training phase. This is achieved via trajectory-relabeling, wherein a trajectory collected for a training task $\psi^i$ is reused or re-purposed for training a different task $\psi^j$. Reward-based trajectory-relabeling has received a lot of attention in recent works on multi-task RL and goal-conditioned RL (Andrychowicz et al., 2017; Eysenbach et al., 2020; Li et al., 2020). The intuition is that if a trajectory $\tau$ collected while solving for the task $\psi^i$ achieves high returns under the reward definition for another task $\psi^j$ (*i.e.,* $\sum_t r_{\psi^j}(s_t, a_t)$ is large), then $\tau$ can be readily used for policy-optimization for the task $\psi^j$ as well. The meta-RL setting presents the following subtlety – for any given task, the meta-RL agent generates trajectories with the aim of utilizing them in the adaptation procedure and subsequently seeks to maximize the post-adaptation returns (*cf.* §3.2). To improve the efficiency of the meta-training stage, we would like to share these pre-adaptation trajectories amongst the different tasks, accounting for the fact that the metric of interest with these trajectories is their usefulness for task-identification, rather than the returns (as in multi-task RL). This difference is illustrated in Figure 3. Hence, when deciding if a trajectory $\tau$ collected for task $\psi^i$ is appropriate to be reused for task $\psi^j$, it is sub-optimal to consider the return value of this trajectory under $\psi^j$. Instead, we argue that this reuse compatibility should be determined based on the performance on the task $\psi^j$, *after* the agent has undergone adaptation using $\tau$. Concretely, we define a function to measure the utility of the trajectory $\tau$ for a task $\psi^j$:

$$U_{\psi^j}(\tau) = \mathbb{E}_{s_t, a_t \sim \pi'} \left[ \sum_{t=1}^{\infty} \gamma^{t-1} r_{\psi^j}(s_t, a_t) \right] \quad (4)$$

where $\pi' = f_\phi\left(\pi_\theta, \tau, r_{\psi^j}\right)$ denotes the policy after using $\tau$ for adaptation. The trajectory-relabeling mechanism during meta-training now incorporates this function $U_{\psi^j}$, which we refer to as the *utility function*, rather than the return $R_{\psi^j}$. Broadly, a trajectory $\tau$ collected for task $\psi^i$ can be relabeled for use in another task $\psi^j$ if $U_{\psi^j}(\tau)$ is high. Subsection §4.1 makes this more precise by deriving a relabeling distribution $q(\psi|\tau)$ that informs us of the tasks for which $\tau$ should be reused. Figure 4, and the caption therein, describe a high-level overview of our approach, HFR.

**Comparison to HIPI (Eysenbach et al., 2020) with a didactic example.** We consider a toy environment to further motivate that return-value based data sharing and trajectory relabeling (as proposed by HIPI) is potentially sub-optimal for meta-RL. The Four-Corners environment consists of a point robot placed at the center of a square where each corner of the square represents a goal location, as shown in Figure 1. For each goal (task), there is a section of the space in the corresponding quadrant in which the robot receives a large negative reward. Consider a trajectory $\tau$ that hovers over the blue square in top-right quadrant. Note that $\tau$ could have been generated by the agent while collecting data for any of the four tasks. We examine if $\tau$ can be reused for the blue task. Since $R_{\text{blue}}(\tau)$ is highly negative, the relabeling strategy in HIPI does not reuse $\tau$ for meta-training on the blue task. It is clearly evident, however, that $\tau$ carries a significant amount of signal pertaining to task-identification on the blue task, making it a useful pre-adaptation trajectory. HFR reuses $\tau$ for the blue task since the utility $U_{\text{blue}}(\tau)$ is high.

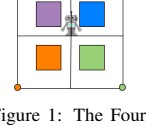

Figure 1: The Four-Corners environment

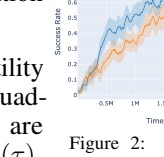

Figure 2: Success-rate on Four-Corners

To quantify this effect, we include the numerical data on the returns and the utility values for a sampled trajectory that hovers over the blue square in top-right quadrant. The values for the returns $\{R_{\text{purple}}(\tau), R_{\text{blue}}(\tau), R_{\text{orange}}(\tau), R_{\text{green}}(\tau)\}$ are $\{-20, -\mathbf{58}, -20, -20\}$, while the utility values $\{U_{\text{purple}}(\tau), U_{\text{blue}}(\tau), U_{\text{orange}}(\tau), U_{\text{green}}(\tau)\}$ are $\{-1015, -\mathbf{756}, -935, -931\}$. These (unnormalized) numbers show that the probability of relabeling this trajectory with the blue task is low under HIPI, but high under HFR. Further analysis is included in Appendix A.6. Figure 2 compares HFR and HIPI in terms of the success-rate in the Four-Corners environment, and shows the performance benefit of using the utility function for trajectory relabeling.

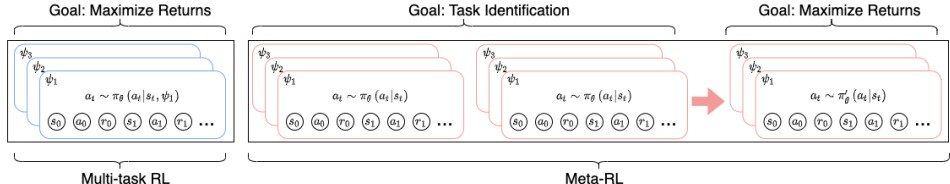

Figure 3: An illustration of the differences between multi-task RL and meta-RL. In multi-task RL (blue) the agent simply maximizes its returns given a task $\psi$, while in meta-RL (orange) the agent must first quickly identify the task with a limited number of exploratory trajectories (first two orange stacks in the figure), before adapting to the task and maximizing returns. Because of these differences, existing multi-task relabeling methods may be sub-optimal for meta-RL.

### 4.1 DERIVING A META-RL RELABELING DISTRIBUTION

Our derivation in this subsection largely follows HIPI (Eysenbach et al., 2020), but differs in that we adapt it to the meta-RL setting to promote sharing of pre-adaptation trajectories amongst tasks, using the concept of trajectory utility. Assume a dataset $\mathcal{D}$ of trajectories gathered by the meta-RL agent when solving the different tasks in the meta-train task distribution. We wish to learn a trajectory relabeling distribution $q(\psi|\tau)$ such that, given any trajectory $\tau \sim \mathcal{D}$, we could reuse $\tau$ for tasks with high density under this posterior distribution. To that end, we start by defining a variational distribution $q(\tau|\psi)$ to designate the trajectories used for the adaptation process $f_\phi$, for a given task $\psi$. Using the definition of the utility function (Eq. 4), the meta-RL objective from Eq. 1 could be written as: $\max_{\theta,\phi} \mathbb{E}_{\psi \sim p(\psi)} \mathbb{E}_{\tau \sim q(\tau|\psi)}[U_\psi(\tau)]$. For fixed meta-parameters $(\theta, \phi)$, a natural approach to optimize the variational distribution $q(\tau|\psi)$ is to use this same objective since it facilitates alignment with the goals of the meta-learner. Thus, the combined objective for the variational distributions for *all* the tasks, augmented with entropy regularization, is:

$$\max_q \mathbb{E}_{\psi \sim p(\psi)} \left[ \mathbb{E}_{\tau \sim q(\tau|\psi)}[U_\psi(\tau)] + \mathcal{H}_{q(\tau|\psi)} \right] \tag{5}$$

where $\mathcal{H}_{q(\tau|\psi)}$ denotes the causal entropy of the policy associated with $q(\tau|\psi)$. Now, we note that the above optimization is equivalent to a reverse-KL divergence minimization objective: $\min_{q(\tau,\psi)} D_{\mathrm{KL}}\left[q(\tau,\psi) \| p(\tau,\psi)\right]$ (*cf.* Appendix A.10). Here, the joint distributions over the tasks and the trajectories are defined as $q(\tau, \psi) = q(\tau|\psi)p(\psi)$ and $p(\tau, \psi) = p(\tau|\psi)p(\psi)$, where

$$p(\tau|\psi) \triangleq \frac{1}{Z(\psi)} p_1(s_1) e^{U_\psi(\tau)} \prod_{t=1}^{T} \mathsf{p}(s_{t+1}|s_t, a_t) \tag{6}$$

Our goal is to formulate the trajectory relabeling distribution $q(\psi|\tau)$. To make this explicit in our objective, we use the trick proposed in HIPI (Eysenbach et al., 2020) and factor $q(\tau, \psi)$ as $q(\psi|\tau) q(\tau)$, thereby rewriting the reverse-KL divergence minimization objective as:

$$\min_{q(\tau,\psi)} \mathbb{E}_{\substack{\tau \sim q(\tau) \\ \psi \sim q(\psi|\tau)}} \left[ \log q(\psi|\tau) + \log q(\tau) - \log p(\psi) + \log Z(\psi) - U_\psi(\tau) - \log p_1(s_1) - \sum_t \log \mathsf{p}(s_{t+1}|s_t, a_t) \right] \tag{7}$$

Ignoring the terms independent of $\psi$, we can analytically solve (by differentiating and setting to zero) for the *optimal* trajectory relabeling distribution for the meta-RL setting:

$$q(\psi|\tau) \propto p(\psi) e^{U_\psi(\tau) - \log Z(\psi)} \tag{8}$$

Given a trajectory $\tau \sim \mathcal{D}$, the HFR algorithm uses this relabeling distribution to sample tasks for which $\tau$ should be reused. Concretely, we compute the utility function under $\tau$ for all the tasks, construct the distribution $q(\psi|\tau)$ using these utilities (Eq. 8), and sample tasks from it. Please see Figure 4 for details. We assume a uniform prior $p(\psi)$ over the tasks in our experiments.

### 4.2 ALGORITHM AND IMPLEMENTATION DETAILS

Our relabeling algorithm is summarized in Algorithm 1 and fits seamlessly into the meta-training process of any of the base meta-RL algorithms. Once the meta-RL agent generates a trajectory $\tau$ for a training task, $\tau$ is fed as input to HFR, and it returns another task that can reuse this experience $\tau$.

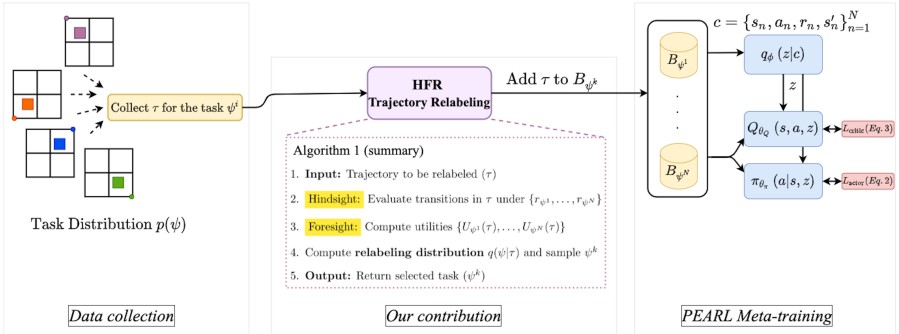

Figure 4: During meta-training, after a trajectory $\tau$ is collected for task $\psi^i$, HFR uses *hindsight* to relabel this trajectory using reward functions for different tasks, and then uses *foresight* to compute the utility of the relabeled trajectory for the different tasks. A distribution over tasks is constructed using the utilities, and a task $\psi^k$ is sampled from the distribution, with tasks for which the trajectory has higher (normalized) utility having higher probability mass. The trajectory is then relabeled using the reward function $r_{\psi^k}$ and added to the task-specific replay buffer $B_{\psi^k}$. Finally, the meta-training update rules are applied. This process repeats throughout the entirety of meta-training. HFR uses PEARL as the base meta-RL algorithm and does not alter its data collection or meta-gradient computation rules. Please see the Algorithm 1 box for details.

---

**Algorithm 1:** Hindsight Foresight Relabeling (HFR)

**Input** : Trajectory to be relabeled ($\tau$)
**Output** : Task to relabel the trajectory with ($\psi$)

**for** *each training task $\psi^i$* **do**
    $U_{\psi^i}(\tau) \leftarrow$ ComputeUtility$(\tau, \psi^i)$
    $\log Z(\psi^i) \leftarrow$ GetLogPartition$(\psi^i)$
**end**
Return $\psi \sim$ softmax$\{U_{\psi^i}(\tau) - \log Z(\psi^i)\}$   (Eq. 8)

**Function** GetLogPartition$(\psi)$:
    Sample batch of trajectories $\{\tau^i\}_{i=1}^N \sim B_\psi$
    **for** *each trajectory $\tau^i$* **do**
        $U_\psi(\tau^i) \leftarrow$ ComputeUtility$(\tau^i, \psi)$
    **end**
    Return $\log Z(\psi) \approx \log\left(\frac{1}{N}\sum_{i=1}^N e^{U_\psi(\tau^i)}\right)$

**Function** ComputeUtility$(\tau, \psi)$:
    **for** *each $(s_t, a_t, r_t) \in \tau$* **do**
        Replace $r_t$ with $r_\psi(s_t, a_t)$
    **end**
    Sample embedding using encoder $z \sim q_\phi(z|\tau)$
    Sample a batch of initial states $\{s_1^i\}_{i=1}^{N_U} \sim B_\psi$

    Sample actions for these states using the
      post-adaptation policy $\pi_\theta(\cdot|s, z)$:
      $\{a_1 \sim \pi_\theta(a_1^i|s_1^i, z)\}_{i=1}^{N_U}$

    Return $U_\psi = \frac{1}{N_U}\sum_{i=1}^{N_U} Q_\theta(s_1^i, a_1^i, z)$   (Eq. 9)

---

We compute the utility of the input trajectory for every training task, along with an empirical estimate of the log-partition function of the tasks. The task to relabel the trajectory with is then sampled from a categorical distribution. For our experiments, we build on top of the PEARL algorithm (Rakelly et al., 2019), which is a data-efficient off-policy meta-RL method. PEARL maintains task-specific replay buffers $B_\psi$. If HFR returns the task $\psi'$, then $\tau$ is relabeled using the reward function $r_{\psi'}$ and added to $B_{\psi'}$ for meta-training on the task $\psi'$.

The adaptation procedure $\pi' = f_\phi(\pi_\theta, \tau, r_\psi)$ for a task $\psi$ corresponds to a sequence of steps: 1.) augment $\tau$ by marking each transition with a reward value computed using $r_\psi(s_t, a_t)$, 2.) condition the encoder on $\tau$ to sample an embedding, $z \sim q_\phi(z|\tau)$; and 3.) condition the policy on $z$ to obtain the post-adaptation policy, $\pi' = \pi_\theta(\cdot|s, z)$. The calculation of the utility function (Eq. 4) requires generation of post-adaptation trajectories, which could be computationally inefficient, especially if the number of tasks is large. To avoid this cost, for each task, we sample a batch of initial states $s_1 \sim p_1(s_1)$ and the corresponding actions from the post-adaptation policy, and compute the utility based on an estimate of the state-action value function $Q_\psi^{\pi'}(s_1, a_1)$ as:

$$U_\psi(\tau) = \mathbb{E}_{s_1 \sim p_1, a_1 \sim \pi'(\cdot|s_1)}\left[Q_\psi^{\pi'}(s_1, a_1)\right] \tag{9}$$

Since we use PEARL, we can avoid training separate task-specific value functions $Q_\psi$, and instead get the required estimates from the task-conditioned critic $Q(s, a, z)$ already used by PEARL (Eq. 3). We

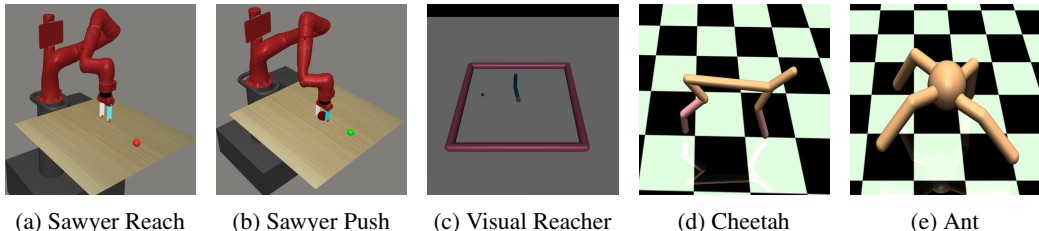

| (a) Sawyer Reach | (b) Sawyer Push | (c) Visual Reacher | (d) Cheetah | (e) Ant |

Figure 5: MuJoCo environments we evaluate on – sparse reward manipulation tasks (a, b, c), as well as sparse and dense reward locomotion tasks (c, d).

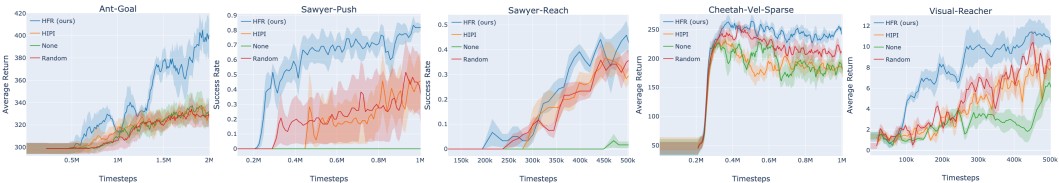

Figure 6: Performance of our relabeling algorithm HFR (shown in blue) on sparse reward tasks. HFR consistently outperforms baselines on both sparse reward robotic manipulation and locomotion tasks. Visual-Reacher uses image observations, while the other environments use proprioceptive states.

highlight that HFR facilitates efficient data-sharing among the training tasks via trajectory-relabeling *without* altering the meta-train and test-time adaptation rules of the base meta-RL algorithm.

## 5 EXPERIMENTS

The goal in this section is to quantitatively evaluate the benefit of sharing experience among tasks using HFR, during the meta-train stage. We evaluate on a set of both sparse and dense reward MuJoCo environments (Todorov et al., 2012) modeled in OpenAI Gym (Brockman et al., 2016). Please refer to the Appendix for environment details. We compare HFR with two relabeling methods: **Random**, in which each trajectory is relabeled with a randomly chosen task, and **HIPI** (Eysenbach et al., 2020), which utilizes MaxEnt RL to devise an algorithm that relabels each transition using a distribution over tasks involving the soft Q values for that transition. In contrast, HFR proposes the concept of the utility of a trajectory for a given task. Using the utility aligns the relabeling methodology with the objective of the meta-RL agent (*cf.* section 4). All methods are built on top of PEARL. Finally, we compare to PEARL with no relabeling at all, which we refer to as **None**.

### 5.1 RESULTS

**Sparse Reward Environments.** We first evaluate HFR on a set of sparse reward robotic manipulation and locomotion tasks. We use five environments: a goal-reaching task involving a quadruped Ant robot, a pushing task on the Sawyer robot, a reaching task on the Sawyer robot, a velocity-matching task involving a bipedal Cheetah robot, and a reaching task involving the MuJoCo Reacher where the agent learns directly from images. The environments are described in detail in Appendix A.4 and shown pictorially in Figure 5. Figure 6 plots the performance (average returns or success-rate) on the held-out meta-test tasks on the $y$-axis, with the total timesteps of environment interaction for meta-training on the $x$-axis. We note that HFR tends to be more sample-efficient than the baselines and achieves a higher asymptotic score. Meta-RL in sparse reward tasks is hard due to the challenges of task-identification and efficient exploration. HFR is especially useful for these tasks as the data-sharing afforded by the trajectory relabeling algorithm mitigates the need for an elaborate exploration strategy during meta-training. This leads to the sample-efficiency gains exhibited in Figure 6.

**Dense Reward Environments.** We next evaluate HFR on dense reward environments (Figure 7). We experiment with the Cheetah-Highdim environment, in which the bipedal robot is required to

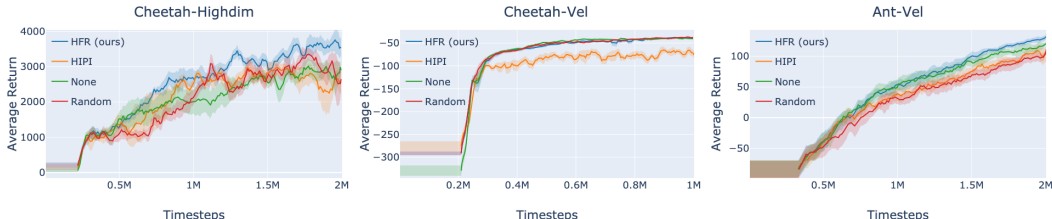

Figure 7: Performance of our relabeling algorithm HFR (shown in blue) on dense reward tasks. With the exception of Cheetah-Highdim, relabeling in general offers no benefit in dense reward meta-RL tasks, likely due to the highly informative nature of a dense reward function.

best match its state vector to a set of predefined vectors, as well as the Cheetah-Vel and quadruped Ant-Vel environments, in which the robots are required to run at various velocities.

Note that the impact of HFR is much less pronounced for these environments, with the exception of Cheetah-Highdim. We believe this is because exploration is not as critical for these environments as it was for the sparse reward tasks. This hypothesis is supported by the fact that, in these environments, PEARL with no relabeling is competitive with the various relabeling methods, which all share similar performance, whereas in the sparse reward environments HFR is the relabeling method that performs best, with the two other relabeling methods also vastly outperforming vanilla PEARL. In the case of the Cheetah-Vel and Ant-Vel environments, agents are provided with an informative dense reward that immediately informs them as to which task they're supposed to be solving. Although the agent in Cheetah-Highdim is also provided with an informative dense reward, the reward function in this task is a linear combination of an 18-dimensional state vector. Thus, reasonably good exploration is needed to determine optimal values for each of these 18 dimensions. HFR relabeling provides improvement over the baselines for this task.

## 5.2 ABLATION STUDIES

**Batch Size.** We investigate the impact of the batch size $N_U$ used in Algorithm 1 for computing an empirical estimate of the state-action value: $\frac{1}{N_U} \sum_{i=1}^{N_U} Q_\theta \left( s_1^i, a_1^i, z \right)$. We compare the effect of batch size across batch sizes $\{16, 32, 64, 128, 256\}$. We expect lower values of $N_U$ to lead to a higher variance estimate of the post-adaptation cumulative discounted rewards and thus potentially a worse approximation of the optimal meta-RL relabeling distribution. In Figure 8a, we see some evidence of this in the comparatively worse performance when using $N_U = 16$ and $N_U = 32$. However, we note that even with these small batch sizes, our method still performs well, and in general achieves high returns across all choices of $N_U$.

**Partition Function.** We investigate the impact of the log-partition function $\log Z(\psi)$ in the optimal meta-RL relabeling distribution (Eq. 8). Prior work has noted the importance of the partition function in the multi-task RL setting when tasks may have different reward scales (Eysenbach et al., 2020). We believe that in the meta-RL setting, the partition function may be crucial even if the tasks share the reward scale, since some tasks in the meta-training distribution may be easier to solve than others. We speculate that with the omission of the partition function from our relabeling distribution, trajectories would find high utility and thus be disproportionately labeled with the easily-solved tasks, causing a degradation in overall performance. In our experiments, we find that the partition function serves as an essential normalization factor; Figure 8b shows an example.

**Reward Function.** One assumption our method assumes is access to the true reward function $r_\psi (s, a)$, which we can query to get the reward for an individual transition under any training task $\psi$. This availability has also been utilized in existing works on relabeling for multi-task RL (Eysenbach et al., 2020; Li et al., 2020). In many real-world applications of meta-RL, *e.g.* a distribution over robotic tasks, it is reasonable to assume that the task-designer outlines a rough template for the rewards corresponding to the different tasks from the distribution. Furthermore, several of the rewards used in our experiments are success/failure indicators, which are simple to specify. Nevertheless, we consider the scenario where we cannot query the true reward function for individual transitions. Figures 8c and 8d show good performance even when we use a learned reward function rather than the true reward function to relabel trajectories.

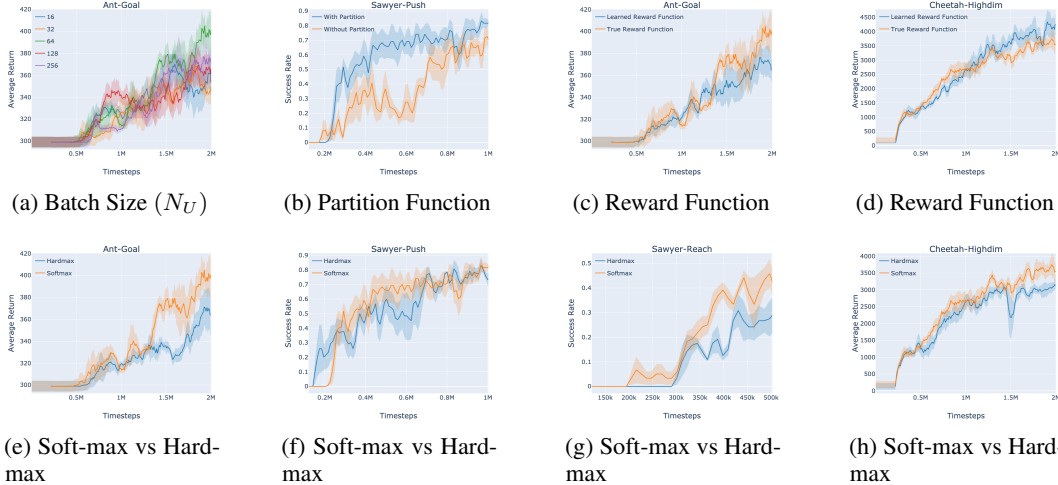

(a) Batch Size ($N_U$)     (b) Partition Function     (c) Reward Function     (d) Reward Function

(e) Soft-max vs Hard-max     (f) Soft-max vs Hard-max     (g) Soft-max vs Hard-max     (h) Soft-max vs Hard-max

Figure 8: Ablation analyses. (a) HFR with different values for the batch size $N_U$ used in approximating Eq. 9. HFR is relatively robust to choice of batch size, with some smaller choices of $N_U$ leading to slightly worse performance. (b) HFR with and without the log-partition function ($\log Z(\psi)$). The partition function serves as a necessary normalization factor and prevents against simply relabeling every trajectory using the easiest task. (c), (d) HFR with true reward function (orange) and HFR with learned reward function (blue). (e), (f), (g), (h) HFR with soft-max (orange) vs hard-max (blue) relabeling distribution

**Soft-max vs Hard-max Relabeling Distribution.** Given a trajectory $\tau$, the relabeling distribution derived in Section §4.1 samples tasks for which $\tau$ should be reused. Specifically, tasks are sampled as: $\psi \sim \texttt{softmax}\{U_{\psi^i}(\tau) - \log Z(\psi^i)\}$. This raises the following question: is it crucial to have stochasticity in the relabeling distribution, or could we deterministically select the task for which the normalized utility value is the highest, *i.e.,* $\psi = \texttt{argmax}\{U_{\psi^i}(\tau) - \log Z(\psi^i)\}$?

A minor modification to the equations in Section §4.1 yields the hard-max relabeling distribution. Concretely, we can add a term to the starting objective for the variational distribution $q$ that explicitly *minimizes* the entropy of the relabeling distribution $q(\psi|\tau)$:

$$\max_q \mathbb{E}_{\psi \sim p(\psi)}\left[\mathbb{E}_{\tau \sim q(\tau|\psi)}[U_\psi(\tau)] + \mathcal{H}_{q(\tau|\psi)}\right] - (1-\epsilon)\mathbb{E}_{\tau \sim q(\tau)}\left[\mathcal{H}_{q(\psi|\tau)}\right]$$

where $\epsilon$ is a value less than 1. Proceeding with the derivation in the exact same manner as in Section §4.1, we obtain the adjusted relabeling distribution:

$$q_\epsilon(\psi|\tau) \propto e^{\frac{U_\psi(\tau) - \log Z(\psi)}{\epsilon}}$$

In the limit when $\epsilon \to 0$, $q_\epsilon(\psi|\tau)$ is the hard-max relabeling distribution. Figures 8e, 8f, 8g, and 8h compare the performance of HFR when sampling tasks from a soft-max relabeling distribution (orange) vs a hard-max distribution (blue). The results indicate that stochasticity is an important factor.

## 6   CONCLUSION

In this paper, we introduced HFR, a trajectory relabeling method for meta-RL that enables data sharing between tasks during meta-train. We argue that unlike the multi-task RL setting, where the appropriateness of a trajectory for a task could be measured by the returns under the task, for meta-RL, it is preferable to consider the future (expected) task-returns of an agent adapted using that trajectory. We capture this notion by defining the *utility* function for a trajectory-task pair and incorporate these utilities in our relabeling mechanism. Inspired by prior work on multi-task RL, an optimal relabeling distribution is then derived that informs us of the tasks for which a generated trajectory should be reused. *Hindsight* is used to relabel trajectories with different reward functions, while *foresight* is used in computing the utility of each trajectory under different tasks and constructing a relabeling distribution. HFR is easy to implement, can be integrated into any existing meta-RL algorithm, and yields improvement on a variety of meta-RL tasks, especially those with sparse rewards. To the best of our knowledge, HFR is the first relabeling method designed explicitly for the meta-RL paradigm.

## 7    REPRODUCIBILITY

We provide our code at https://github.com/michaelwan11/hfr. Relevant hyperparameters can be found in Tables 1 and 2 as well as the config files in our code, while algorithm details can be found in Algorithm 1.

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

## A  APPENDIX

### A.1  FAILED EXPERIMENTS (AN ALTERNATE UTILITY FUNCTION)

While the expected post-adaptation return is a direct indicator of the usefulness of a trajectory for a task, we now mention an indirect measure that is pertinent to the meta-RL baseline algorithm we employ (PEARL). The idea behind PEARL is that the encoder network $q_\phi$, conditioned on the given trajectory (referred to as "context"), produces an embedding $z$ that captures some salient information about the task. The encoder is trained to minimize the Bellman error ($L_{\text{critic}}$, Eq. 3). A lower value of $L_{\text{critic}}$ thus indicates that the trajectory is valuable for identifying the task $\psi$. Therefore, we experiment with using the negative of $L_{\text{critic}}$ as a utility function:

$$U_\psi(\tau) = - \mathbb{E}_{\substack{(s,a,r,s')\sim B_\psi, \\ z \sim q_\phi(z|\tau)}} \left[ \left( Q_\theta(s,a,z) - (r + V(s',z)) \right)^2 \right] \tag{10}$$

We can efficiently compute this utility function with a single forward pass through the encoder, followed by computation of the Bellman error using a batch of $(s, a, r, s')$ tuples. Algorithm 2 includes the details. We note that this utility function explicitly gauges the viability of task-identification through the PEARL encoder, rather than measure the post-adaptation returns.

---

**Algorithm 2:** Computation of the utility function based on the Bellman error (Eq. 10), for PEARL-based meta-RL

---
**Input** : Trajectory to be relabeled ($\tau$)
   Task to compute utility of the trajectory for ($\psi$)
**Output** : Utility of the trajectory $\tau$ under task $\psi$, $U_\psi(\tau)$

**for** *each* $(s_t, a_t, r_t) \in \tau$ **do**
  | Replace $r_t$ with $r_\psi(s_t, a_t)$
**end**
Sample embedding using the encoder $z \sim q_\phi(z|\tau)$
Sample a batch of transitions $\{s_i, a_i, r_i, s_i'\}_{i=1}^N \sim B_\psi$
Return the negative Bellman error (Eq. 10): $U_\psi = \frac{1}{N} \sum_{i=1}^N - (Q_\theta(s_i, a_i, z) - (r_i + V(s_i', z)))^2$

---

Our results when using this new utility function can be seen in Figure 9. Overall, relabeling based on the Bellman error (HFR-Bellman) does not perform as well as relabeling based on the expected post-adaptation returns, likely because the Bellman error is simply an indirect measure of the true metric of interest (post-adaptation returns).

### A.2  TRANSITION DYNAMICS

Although this work focuses on the setting where the *tasks share the same transition dynamics, but differ in the reward function*, in this section we show results on environments where the tasks differ in *both the transition dynamics and reward function*. We consider two environments: **Ant-Goal-Rand-Params** and **Cheetah-Vel-Sparse-Rand-Params**, which are identical to **Ant-Goal** and **Cheetah-Vel-Sparse**, respectively, except for the fact that the transition dynamics now also differ across tasks. We do not modify any of the methods (HFR, HIPI, None, Random) to take into consideration the different transition dynamics across tasks. Our results can be seen in Figure 10. We find that (a) Relabeling can surprisngly still provide some benefit in this setting and (b) HFR still outperforms baselines when transition dynamics differ.

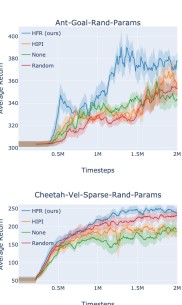

Figure 10: Performance on tasks that differ in both reward function and transition dynamics.

### A.3  HYPERPARAMETERS

Table 1 lists the hyperparameters that were shared across all environments. Hyperparameters for the Sawyer environments were taken from Yu et al. (2020), while hyperparameters for the other environments were taken from the open-source PEARL implementation (Rakelly et al., 2019).

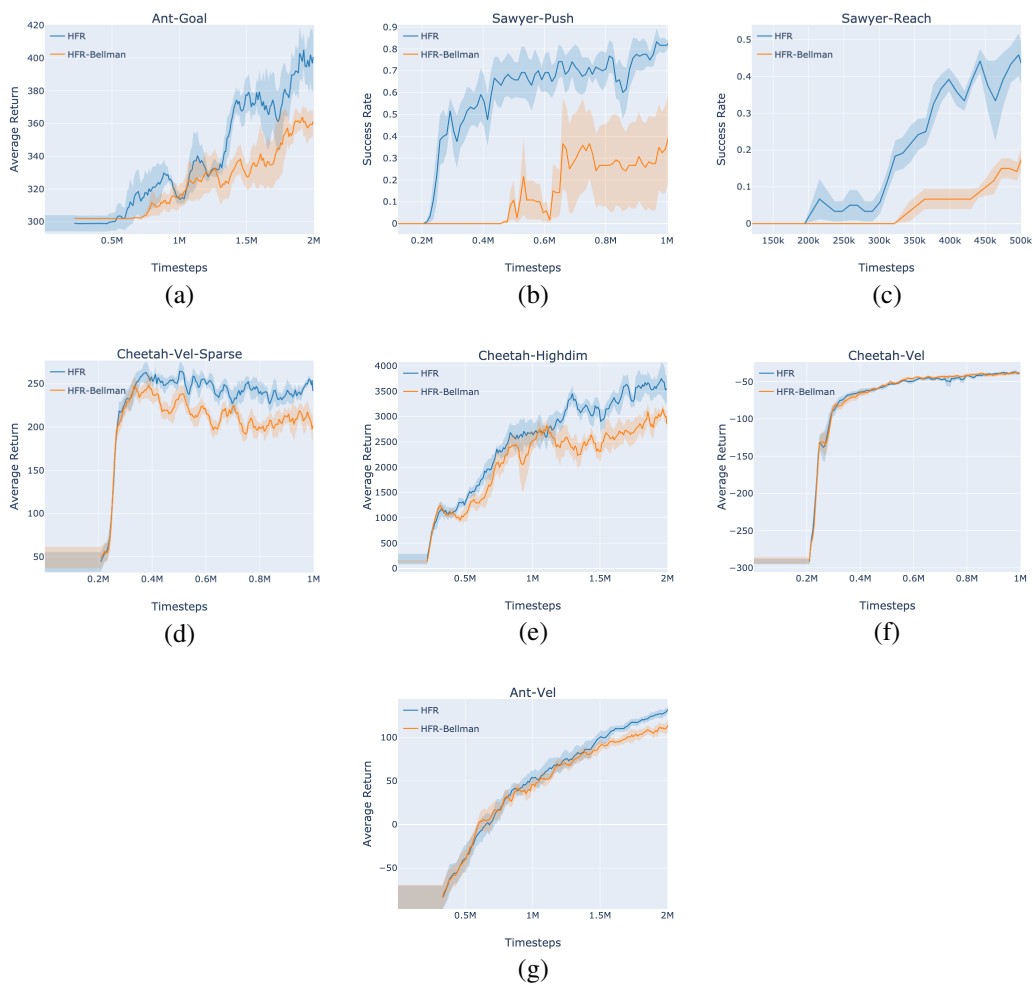

Figure 9: Performance of our relabeling algorithm HFR (blue) compared to a variant of HFR based on the Bellman error (orange)

Table 1: PEARL hypeparameters used for all experiments.

| Hyperparameter | Value |
|---|---|
| Nonlinearity | ReLU |
| Optimizer | Adam |
| Policy Learning Rate | 3e−4 |
| Q-function Learning Rate | 3e−4 |
| Batch Size | 256 |
| Replay Buffer Size | 1e6 |

## A.4 ENVIRONMENTS

The environments we evaluate HFR on are as follows:

**Four-Corners:** We create a 2D navigation task in which a point robot must navigate to one of four goal locations. A reward of $0$ is given when the robot is within a distance of 0.2 from the goal, with the episode ending. Otherwise, a reward of $-1$ is given, with a reward of $-3$ being given if the robot is within a certain section of space in the same quadrant as the goal (Figure 1).

**Ant-Goal:** We use the Ant-Goal task from (Gupta et al., 2018). Tasks correspond to goal locations sampled uniformly from a half-circle of radius 2. The reward function is $4 - ||x_\text{ant} - x_\text{goal}||_2$ if $||x_\text{ant} - x_\text{goal}||_2 \leq 0.8$ and $4 - c$ otherwise, where $c$ is a large uninformative constant.

Table 2: Environment Details

| Environment | Discount | Horizon | Train Tasks | Test Tasks | Number of Exploration Steps |
|---|---|---|---|---|---|
| Ant-Goal | 0.99 | 200 | 100 | 30 | 400 |
| Ant-Vel | 0.99 | 200 | 150 | 30 | 400 |
| Cheetah-Highdim | 0.99 | 200 | 100 | 30 | 400 |
| Cheetah-Vel | 0.99 | 200 | 100 | 30 | 400 |
| Cheetah-Vel-Sparse | 0.99 | 200 | 100 | 30 | 400 |
| Four-Corners | 0.90 | 20 | 4 | 4 | 380 |
| Sawyer-Push | 0.99 | 150 | 50 | 10 | 450 |
| Sawyer-Reach | 0.99 | 150 | 50 | 10 | 450 |
| Visual-Reacher | 0.99 | 100 | 50 | 10 | 200 |

**Ant-Vel:** We use the Ant environment from OpenAI gym. Tasks correspond to goal velocities sampled uniformly in $[0, 3]$. The reward is the negative absolute value of the difference between the agent's velocity and the goal velocity. We take this task from (Finn et al., 2017).

**Cheetah-Highdim:** We take the Cheetah-Highdim task from (Lin et al., 2020). Each task corresponds to an 18-dimensional vector $\psi$ sampled uniformly in $[-1, 1]^{18}$. The reward is linear in the post-transition state $s'$ and is given as $r(s, a, s') = \psi^T s'$.

**Cheetah-Vel:** The Cheetah-Vel task is taken from (Rakelly et al., 2019). Velocities are sampled uniformly in $[0, 3]$ and the reward is the negative absolute value of the difference between the agent's velocity and the goal velocity.

**Cheetah-Vel-Sparse:** Velocities are sampled uniformly in $[0, 3]$, with positive reward being given if the absolute value of the difference between the agent's velocity and the goal velocity is less than $0.3$.

**Sawyer-Push:** An agent must control a simulated Sawyer robot and push a block to a specified goal location. A reward of $0$ is given when the block is within a distance of $0.07$ of the goal and a reward of $-1$ is given otherwise. The episode ends when either the goal is reached or the agent has taken 150 steps. Both the Sawyer Reach and Sawyer Push environment were taken from (Yu et al., 2020).

**Sawyer-Reach:** An agent must control a simulated Sawyer robot and reach a specified 3D goal location. A reward of $0$ is given when the end effector is within a distance of $0.05$ of the goal and a reward of $-1$ is given otherwise. The episode ends when either the goal is reached or the agent has taken 150 steps.

**Visual-Reacher:** We use the standard MuJoCo Reacher environment where an agent must reach various 2D goal locations, with positive reward being given if the end effector is within a distance of $0.03$ of the goal. The agent must meta-learn directly from $64{\times}64$ grayscale images of the environment as the input observations.

Table 2 includes the details for each environment and Figure 5 shows them pictorially. The average return shown in our plots is the average return obtained after N initial exploration steps, where N is the value under Number of Exploration Steps, after which the agent should attempt to solve the task.

### A.5 Time/Space Complexity

HFR incurs an $O(NC)$ cost every time a trajectory is collected, where $N$ is the number of tasks in the training task distribution and $C$ is the cost of computing the utility of a relabeled trajectory. Computing the utility of a relabeled trajectory involves passing this trajectory through the encoder to generate a latent embedding, sampling a batch of initial states from the replay buffer, passing these initial states and embedding to the policy network to generate a batch of actions, and evaluating the Q function on these state-action pairs (Algorithm 1). HFR takes $O(N)$ space to store the relabeling distribution.

### A.6 Analysis of Trajectories in the Four-Corners environment

In this section, we analyze sample trajectories in the Four-Corners environment, as shown in Figure 11. Each trajectory's original goal is different from the goal located in the quadrant it explores. In Table 3,

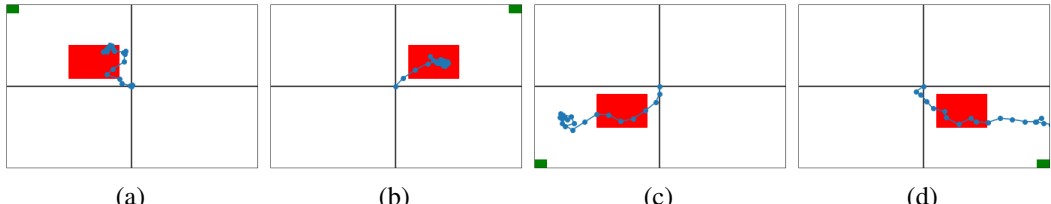

|  | (a) | (b) | (c) | (d) |

Figure 11: Example trajectories from our Four-Corners Environment. The goal for the task drawn from HFR's relabeling distribution is shown in green, while the red square represents an area of large negative reward for that task. Encountering the red square informs the agent of the goal location. It is clear that trajectories (a), (b), (c), and (d) are most useful for meta-training on how to reach the top left goal, top right goal, bottom left goal, and bottom right goal, respectively. HFR correctly relabels these trajectories with the appropriate task, while HIPI fails to do so due to the large negative returns achieved under them. Table 3 shows the trajectory returns that HIPI uses, and the unnormalized Q-values that HFR uses, to sample a task to relabel trajectories (a), (b), (c), and (d) with.

we show trajectory returns under the different tasks for HIPI and the unnormalized post-adaptation Q-values under the different tasks for HFR. We note that for the first trajectory, HIPI fails to relabel it using the top left goal due to the highly negative return (-44) it achieves for that goal. However, HFR, by considering the *post-adaptation returns* after using the trajectory for adaptation, correctly relabels the trajectory for the top left goal (-727.46 is the highest value in the column). This is beneficial, as despite the highly negative return, the trajectory is extremely informative about the top left goal and should be used for meta-training on it. The same phenomenon is seen for the other 3 trajectories.

|  | Task | HFR | HIPI |
|---|---|---|---|
| | **Top Left** | $-727.46$ | $-44$ |
| (a) | Top Right | $-957.61$ | $-20$ |
| | Bottom Left | $-985.34$ | $-20$ |
| | Bottom Right | $-937.60$ | $-20$ |

|  | Task | HFR | HIPI |
|---|---|---|---|
| | Top Left | $-1015.28$ | $-20$ |
| (b) | **Top Right** | $-756.99$ | $-58$ |
| | Bottom Left | $-935.45$ | $-20$ |
| | Bottom Right | $-931.33$ | $-20$ |

|  | Task | HFR | HIPI |
|---|---|---|---|
| | Top Left | $-916.60$ | $-20$ |
| (c) | Top Right | $-790.87$ | $-20$ |
| | **Bottom Left** | $-780.03$ | $-28$ |
| | Bottom Right | $-829.24$ | $-20$ |

|  | Task | HFR | HIPI |
|---|---|---|---|
| | Top Left | $-1243.47$ | $-20$ |
| (d) | Top Right | $-1202.90$ | $-20$ |
| | Bottom Left | $-1249.10$ | $-20$ |
| | **Bottom Right** | $-760.03$ | $-30$ |

Table 3: Unnormalized trajectory returns (HIPI) and post-adaptation Q-values (HFR) for trajectories (a), (b), (c), (d) (Figure 11). The task for which the trajectory is most informative is in bold. Given task $\psi$ and trajectory $\tau$, HIPI will not relabel trajectory $\tau$ with task $\psi$ if the return of $\tau$ is low under $\psi$, even though $\tau$ may be extremely useful for meta-training on $\psi$. By considering *post-adaptation returns*, HFR does not suffer from this issue.

## A.7 LEARNING A REWARD FUNCTION

In this section, we describe how we learn a reward function in the setting where we do not assume we can query the true reward function $r_\psi (s_t, a_t)$ for individual transitions, as explored in Subsection §5.2. We can use a learned reward function $r_\psi^\omega (s_t, a_t)$ parameterized by a neural network with parameters $\omega$ to relabel trajectories. This network takes in task $\psi$ and state and action $s_t, a_t$ as input. We train our reward function using the mean-squared error:

$$\min_\omega \mathbb{E}_{\psi \sim p(\psi), s_t, a_t \sim B_\psi} \left[ \left( r_\psi^\omega (s_t, a_t) - r_\psi (s_t, a_t) \right)^2 \right] \tag{11}$$

This update rule is applied during every training update during the meta-training process.

In Algorithm 3, we include the meta-test time procedure of PEARL used in HFR. In PEARL, adaptation procedure $f_\phi$ corresponds to a forward pass through encoder $q_\phi$.

---

**Algorithm 3:** PEARL Meta-testing

---

**Input**   :Test task($\psi$)
        Number of exploration trajectories ($K$)

Initialize context $c_\psi = \{\}$
**for** $k = 1$ **to** $K$ **do**
    Sample embedding $z \sim q_\phi\left(z|c_\psi\right)$
    Collect trajectory $\tau_k = \left\{(s_t, a_t, r_t, s_{t+1})\right\}_{t=1}^N$ using policy $\pi_\theta\left(\cdot|s, z\right)$
    Append trajectory $\tau_k$ to context: $c_\psi = c_\psi \cup \tau_k$
**end**

Sample embedding $z \sim q_\phi\left(z|c_\psi\right)$
Compute adapted policy $\pi' = f_\phi\left(\pi_\theta, c_\psi, r_\psi\right) = \pi_\theta(\cdot|s, z)$
Roll out adapted policy $\pi'$ to measure returns on $\psi$

---

## A.9   IS THE LEARNED Q FUNCTION CORRECTLY CAPTURING TASK INFORMATION?

In this section, we examine whether it makes sense to use the learned Q function as the source of relabeling signal. To do this, we train HFR to convergence, freeze the weights of the HFR context encoder $q_\phi$, and take a set of trajectories $T_N = \left\{\tau_k^N\right\}_{k=1}^K$ from the training process that have not been relabeled.

We split $T_N$ into two collections of trajectories, which we denote as $T_{N_{\text{train}}}$ and $T_{N_{\text{test}}}$. We construct $T_R$ by taking the trajectories from $T_{N_{\text{test}}}$ and relabeling them using HFR (Algorithm 1). We then use our trained HFR context encoder $q_\phi$ to generate three collections of context embeddings: $Z_{N_{\text{train}}} = \left\{z \sim q_\phi\left(\cdot|\tau\right) : \tau \in T_{N_{\text{train}}}\right\}, Z_{N_{\text{test}}} = \left\{z \sim q_\phi\left(\cdot|\tau\right) : \tau \in T_{N_{\text{test}}}\right\}, Z_R = \left\{z \sim q_\phi\left(\cdot|\tau\right) : \tau \in T_R\right\}$.

We then train a $N$-way classifier $f\left(\psi|z\right)$ on $Z_{N_{\text{train}}}$. Given context embedding $z$, $f$ should correctly identify the task $\psi$ that $z$ was generated from. If $z$ was generated from a relabeled trajectory, then $f$ should classify $z$ as coming from the task that the trajectory was relabeled with. If $z$ was generated from a trajectory that was not relabeled, then $f$ should classify $z$ as coming from the task that the trajectory was originally generated for. We evaluate the accuracy of our classifier on $Z_{N_{\text{test}}}$ and $Z_R$. If HFR is correctly capturing task information during relabeling, we would expect the accuracy on $Z_R$ to be greater than or equal to the accuracy on $Z_{N_{\text{test}}}$. For clarity, we illustrate this entire procedure in Algorithm 4.

---

**Algorithm 4:** Context Embedding Classification

---

**Input**   :Trained context encoder($q_\phi$)
        Trajectories($T_N$)
**Output**:Accuracy of classifier on original trajectories
        Accuracy of classifier on relabeled trajectories

Split $T_N$ into $T_{N_{\text{train}}}$ and $T_{N_{\text{test}}}$
Construct $T_R$ by relabeling trajectories in $T_{N_{\text{test}}}$ using HFR (Algorithm 1)

Generate $Z_{N_{\text{train}}} = \left\{z \sim q_\phi\left(\cdot|\tau\right) : \tau \in T_{N_{\text{train}}}\right\}$
Generate $Z_{N_{\text{test}}} = \left\{z \sim q_\phi\left(\cdot|\tau\right) : \tau \in T_{N_{\text{test}}}\right\}$
Generate $Z_R = \left\{z \sim q_\phi\left(\cdot|\tau\right) : \tau \in T_R\right\}$

Train classifier $f\left(\psi|z\right)$ on embeddings from $Z_{N_{\text{train}}}$
Return accuracy of $f$ on $Z_{N_{\text{test}}}$, accuracy of $f$ on $Z_R$

---

We conduct this experiment on the Four-Corners environment, as well as the Cheetah-Vel-Sparse environment with 10 training tasks. We parameterize $f$ as a feed-forward neural network with two hidden layers, each with 200 units, and ReLU activations. We train $f$ using cross-entropy loss for 500 epochs on a training set of non-relabeled trajectories and report the accuracy on relabeled trajectories and a test set of non-relabeled trajectories at the end of training. Our results are shown in Table 4. We find that the accuracy on relabeled trajectories is higher than the accuracy on non-relabeled

Table 4: Accuracy of classifier on relabeled and non-relabeled trajectories

| Environment | Accuracy on Relabeled Trajectories | Accuracy on Non-Relabeled Trajectories |
|---|---|---|
| Four-Corners | 89.65% | 76.98% |
| Cheetah-Vel-Sparse | 74.33% | 57.58% |

trajectories. This indicates that HFR is indeed correctly capturing task information when relabeling – our relabeled trajectories are easier to identify as coming from a specific task than the non-relabeled trajectories and are thus more useful for adaptation.

### A.10    OBJECTIVE EQUIVALENCE

In this section, we show that the objective in Equation 5 is equivalent to the reverse-KL divergence minimization objective: $\min_{q(\tau,\psi)} D_{\text{KL}}\left[q\left(\tau,\psi\right)||p\left(\tau,\psi\right)\right]$, where $q\left(\tau,\psi\right) = q(\tau|\psi)p(\psi)$, and $p\left(\tau,\psi\right) = p(\tau|\psi)p(\psi)$. Further, $p(\tau|\psi)$ is defined in Equation 6.

$$
\begin{aligned}
\min_q D_{\text{KL}}\left[q\left(\tau,\psi\right)||p\left(\tau,\psi\right)\right] &= \min_q \mathbb{E}_{q(\tau,\psi)} \log \frac{q(\tau,\psi)}{p(\tau,\psi)} \\
&= \min_q \mathbb{E}_{p(\psi)q(\tau|\psi)} \log \frac{q(\tau|\psi)}{p(\tau|\psi)} \\
&= \min_q \mathbb{E}_{p(\psi)q(\tau|\psi)} \log \frac{\prod_{t=1}^{T} \tilde{q}(a_t|s_t;\psi)}{e^{U_\psi(\tau)}} \\
&= \max_q \mathbb{E}_{\psi \sim p(\psi)} \left[\mathbb{E}_{\tau \sim q(\tau|\psi)}[U_\psi(\tau)] + \mathcal{H}_{q(\tau|\psi)}\right]
\end{aligned}
$$

where $\mathcal{H}_{q(\tau|\psi)}$ denotes the causal entropy of the policy associated with $q(\tau|\psi)$, that is $\tilde{q}(a|s;\psi)$.

### A.11    COMPARISON TO PRIOR WORK IN EXPERIENCE RELABELING

#### A.11.1    COMPARISON WITH MIER (MENDONCA ET AL., 2020)

Mendonca et al. (2020) propose to tackle meta-RL via a model identification process, where context-dependent neural networks parameterize the transition dynamics and the rewards function. These models are trained with supervised meta-learning. For a new task (at meta-train/test time), the context is adapted to encapsulate the task information. Although MIER uses experience relabeling, the scope and the purpose of this relabeling is very different from HFR:

- **Relabeling during meta-train vs meta-test.** In MIER, experience relabeling is performed only at meta-test time, with the purpose of consistent adaptation to the out-of-distribution tasks. Crucially, there is no relabeling or sharing of data amongst tasks during the meta-train time. In contrast, the motivation of the relabeling in HFR is to improve the sample-efficiency of the meta-training phase. HFR does not modify the meta-test time behavior of the underlying meta-RL algorithm.

- **Problem setup.** MIER considers a setup where the meta-test tasks are "out-of-distribution" *w.r.t.* the meta-train task distribution. The authors, therefore, introduce relabeling to handle this discrepancy. HFR operates in the standard meta-RL setting where the meta-test tasks are "in-distribution". MIER does not perform relabeling in this case (please see Figure 2 in their paper).

- **Relabeling strategy.** Lastly, the relabeling strategy is very different for the two approaches. While MIER relabels *all* of the meta-training data for the new meta-test time task, HFR selectively relabels the trajectory for a new task during meta-train, basing the selection on the idea of the utility function and a relabeling distribution $q(\psi|\tau)$.

#### A.11.2    COMPARISON WITH BOReL (DORFMAN ET AL., 2020)

Dorfman et al. (2020) propose a method for offline meta-RL where the meta-learning agent is provided with fixed trajectory data collected in different environments and the goal of the agent is to

learn to maximize returns in an unseen environment from the same task distribution. The authors highlight the issue of "MDP ambiguity", which refers to the difficulty in MDP identification using the belief from a VAE encoder trained with offline data. To mitigate this challenge, the authors propose a reward relabeling scheme that replaces the reward in a trajectory from an MDP-$i$ in the offline data, with rewards from a randomly chosen MDP-$j$. There are two critical differences from our work:

- **Motivation for relabeling.** Relabeling in BOReL is done to alleviate the "MDP ambiguity" issue, which the authors mention is specific to the offline meta-RL setting. HFR operates in the online meta-RL setting and our motivation for relabeling is to improve the sample-efficiency of the online meta-training phase via data sharing.

- **Relabeling strategy.** Perhaps the more important difference is in the strategy used to select the task for which a given trajectory should be relabeled. While BOReL chooses the task at random, HFR provides a principled procedure of computing a relabeling distribution $q(\psi|\tau)$ over the tasks using the concept of the utility function, and sampling a task from this distribution for the relabeling. Empirically, we observe that this approach is more performant than random task relabeling. Figure 6 in the paper includes the learning curves for both the methods – HFR outperforms Random in all those tasks.

