# OpenReview forum: "Hindsight Foresight Relabeling for Meta-Reinforcement Learning"
_ICLR.cc/2022/Conference — ICLR 2022 Poster_

### Official Review · Reviewer_vofF · 2021-11-02

**Correctness:** 4
**Technical Novelty And Significance:** 3
**Empirical Novelty And Significance:** 3
**Recommendation:** 5
**Confidence:** 3

**Main Review:**

(a). I am a little concerned with the novelty of the paper.  The authors made an interesting point that compared to multi-task RL, the objective in meta-RL is to learn to learn a new task, so the metric of interest for relabeling trajectories in meta-RL is their usefulness for task-identification, rather than the returns like in multi-task RL (Section 4, page 4). The paper’s main contribution is a trajectory-relabeling algorithm in meta-RL setting based on this intuition. However, the proposed algorithm still seems to try to relabel trajectories based on their returns on other tasks. Specifically, the resulting learning objective Equation (8) is quite similar to that of HIPI (Eysenbach et al., 2020), as the utility function similarly aims to maximize the expected return of the trajectory on the new task, which I think would be exactly the total reward of trajectory on the new task in the multi-task setting. This seems to be contradictory to what the papers says about the difference of relabeling trajectories of previous work in multi-task rl and relabeling trajectories in meta-RL proposed in this work. Plus, the actual implementation looks like a straightforward combination of HIPI and PEARL (Rakelly et al., 2019) to me.


(b). The proposed algorithm makes an important assumption that the actual reward function is known for each task in this meta-RL setting. This makes the meta-RL problem setting confusing as the paper also says the tasks share the same dynamics and only differ in the reward function. But like the paper mentions, there may exist some scenarios where this assumption is reasonable. The experimental results show that the proposed algorithm improves performance compared with other relabeling methods (HIPI and random) in such settings. First, I think there might exist methods that, under the same assumptions, are able to do better than meta-RL algorithms. For instance, one can relabel all the collected data with the new task reward and run some kind of offline RL algorithms on it (without meta-learning). In my opinion, that would be another good baseline to strengthen the author’s claim under the same assumptions. Secondly, in the experiment section the paper mentions a variant of the proposed method that also considers the scenario where the true reward function cannot be queried for individual transitions. This is a more interesting setting and I think the authors should elaborate on this part (e.g. how do you learn the reward functions? Is there anything specifically designed for meta-RL settings?) And it would be better to show more experimental results under these settings. For instance, the author can compare to the state-of-the-art meta-RL algorithms under such more common settings, especially in sparse-reward environments where the proposed algorithm is more competent. Potential baseline: 1. MetaCURE: Meta Reinforcement Learning with Empowerment-Driven Exploration, Zhang et al., ICML 2021;  3.  Towards Effective Context for Meta-Reinforcement Learning: an Approach based on Contrastive Learning, Fu et al., AAAI 2021.


(c). I have some minor comments listed below:

1. Equation (4), On the left hand side, $j$ should be superscript instead of subscript.

2. Figure 4, the actor $\pi$ and critic $Q$ should be parameterized using different denotations, instead of jointly using $\theta$.

3. Figure 6, could the authors explain why randomly relabeling the trajectories can achieve competitive performance or even better performance than HIPI and PEARL?

4. The adaptation procedures listed on page 6 before equation (9) confuse me. Could the authors provide an algorithm bar for the meta-test phase (maybe in appendix)?


**Summary Of The Paper:**

The paper proposes a trajectory relabeling method for meta Reinforcement Learning (meta-RL), aiming to share some of the collected trajectories to improve sample efficiency during meta-training. The relabeling method is built on HIPI (Eysenbach et al., 2020). Instead of relabeling the trajectory based on the total reward as in HIPI, the paper argues that in meta-RL, the metric of interest for trajectories from different tasks is their usefulness for task-identification rather than returns. The paper further proposes a meta-RL algorithm based on PEARL (Rakelly et al. 2019). The experimental results on several sparse-reward tasks show that the method outperforms other relabeling methods as well as PEARL.

**Summary Of The Review:**

The idea in the paper is well presented and carefully investigated. The proposed method is simple and effective. However, I am not quite convinced about the novelty of the proposed idea and I think the experimental settings can be improved to strengthen the paper’s claim.

---

> ### Author Response · Authors · 2021-11-21
> **Response to the Reviewer vofF**
>
> Thank you for your thoughtful comments and suggestions! We address the reviewer’s concerns below.
>
> **Concerns about novelty; Equation (8) seems contradictory to claims:** We would like to clear some misunderstanding here. The reviewer writes that _“as the utility function similarly aims to maximize the expected return of the trajectory on the new task, which I think would be exactly the total reward of trajectory on the new task in the multi-task setting.”_. We respectfully disagree with the reviewer since this is an inaccurate claim. HIPI (for multi-task RL) relabels a trajectory based on the return it achieves under various tasks, whereas HFR (for meta-RL) relabels a trajectory based on the expected returns for a specific task after the trajectory has been used for adaptation to that task. **Note that the latter (HFR case) is not equivalent to the return of the trajectory under the new task**. Please see the “Comparison to HIPI” discussion in Section 4, as well as Algorithm 1 in Section 4.2. We do compare our method to a combination of HIPI and PEARL (the HIPI baseline) in our empirical evaluations (Figure 6) and find that our method generally performs better (HFR, as well as our baselines, are all built on top of the PEARL algorithm, with the only difference being the type of relabeling).
>
>
>
> **Alternate methods (e.g. offline-RL) could do better than meta-RL under same assumptions:** We would like to emphasize that we assume that we can query the true reward function for individual transitions _only for the training tasks_. Crucially though, **we do not assume we can query the true reward function for individual transitions for the test tasks**. So it is certainly true, as the reviewer points out, that we should expect offline RL to do well on the training tasks if we have access to the true reward function, as we can generate a large amount of training data for each task through relabeling. However, in meta-RL, we care primarily about performance on the held-out test tasks, where we do not have access to the true reward function. Therefore, offline RL on these test tasks would be infeasible due to the lack of sufficient training data. Furthermore, offline-RL would learn a separate agent for each task -- this solution does not scale elegantly as the task distribution becomes larger. In contrast, meta-RL seeks a single meta-learned agent that could be adapted efficiently to a new task given limited data.
>
>
> **Figure 6: explain why randomly relabeling the trajectories can achieve competitive performance or even better performance than HIPI and PEARL**: We believe Random-relabeling does better than PEARL-no-relabeling due to the exploration challenge present in these sparse reward tasks. Random-relabeling, while not as performant as HFR, can partially alleviate the exploration challenge by providing informative trajectories for the various tasks, though only with a chance probability. The claim is further supported by our dense reward experiments in Figure 7, where PEARL-no-relabeling and Random-relabeling perform similarly in environments where exploration is not as critical.
>
> We believe Random-relabeling is competitive with HIPI in the meta-RL setting since HIPI was designed specifically for multi-task RL, and thus its relabeling suggestions are typically not very effective for meta-RL. The meta-RL setting has some subtle but important differences, as explained in the “Comparison to HIPI'' discussion in Section 4 and Figure 3.
>
> **Details about learned reward function:** Thank you for this suggestion! We have added Appendix A.7 to include these details.
>
> **Minor comments: Equation 4 and Figure 4**: Thank you for the corrections!
>
> **Could the authors provide an algorithm bar for the meta-test phase?** We have added an algorithm bar for the meta-test phase in Appendix A.8, and we hope this clears up any confusion. Please let us know if you still have questions regarding the meta-test phase.

---

> > ### Comment · Reviewer_vofF · 2021-11-27
> > **Response to author rebuttal**
> >
> > I thank the authors for their detailed responses.
> >
> > The meta-test setting that the true reward function is not known is reasonable and partially alleviates my concerns about the problem settings and the possible offline-RL baseline, though knowing each task’s reward function during meta-training still seems weird to me in meta-learning problems that different tasks actually only vary in reward functions. The further explanation about learned reward function in appendix is helpful, while I believe more empirical results of using such learned reward function to relabel and comparison to other standard meta-RL algorithms would greatly strengthen the paper.
> >
> > I appreciate the clarification regarding the difference between HFR and HIPI’s relabeling objective. However, I still believe the main novelty of this paper comes from a straightforward way of adapting the relabeling strategy in multi-task rl to a new problem setting, which is relatively minor. My comment about novelty was that such a change in objective function is straightforward as the problem setting changes from multi-task learning to meta learning, as we care more about the agent’s performance on test tasks instead of all training tasks, which is not a novel insight. Despite this, I do find the results the authors provided in response to reviewer HrwG are convincing and makes the empirical evaluation part of the paper stronger.

---

> > > ### Author Response · Authors · 2021-11-27
> > > **Thank you for your response**
> > >
> > > We really appreciate the reviewer for taking our rebuttal into account and raising their rating of the paper. Thank you! As suggested by the reviewer, we would add empirical results for relabeling with the learned reward function for *all the environments* considered in this paper, in the final revision.

---

> > > ### Author Response · Authors · 2021-11-29
> > > **Further response to the reviewer vofF**
> > >
> > > We’d like to thank the reviewer once again for their valuable feedback. We hope to address the reviewer’s concern about the problem setting below:
> > >
> > > **Knowing each task’s reward function during meta-training:** This assumption that we can query the reward function for individual transitions for each training task has been made in prior works in relabeling for multi-task RL [1, 2]. While the goals of multi-task RL and meta-RL are different (the goal of multi-task RL is to train an agent to accomplish various tasks when given the task identities while the goal of meta-RL is to train an agent to quickly adapt to and solve new tasks), in our view this should not influence whether or not we can query the reward function for the training tasks, as in the end both multi-task RL and meta-RL involve training on a distribution of tasks. Since knowledge of the reward function has been assumed in multi-task RL, we believe that this assumption can also be made in the meta-RL setting. Furthermore, many of our reward functions are simply success/failure indicator functions or extremely simple sparse reward functions which are easy to specify, and usually in practice the task-designer who specifies the meta-training task distribution will have knowledge of the reward function. We hope this alleviates the concerns the reviewer has about the problem setting.
> > >
> > > [1] Rewriting History with Inverse RL: Hindsight Inference for Policy Improvement (Eysenbach et al.)
> > >
> > > [2] Generalized Hindsight for Reinforcement Learning (Li et al.)

---

### Official Review · Reviewer_YEZ4 · 2021-11-03

**Correctness:** 3
**Technical Novelty And Significance:** 2
**Empirical Novelty And Significance:** 3
**Recommendation:** 8
**Confidence:** 3

**Main Review:**

I like this paper overall. The motivation is sound: meta-RL is almost by definition slow, by using a slower timescale for meta-learning than the fast learning or adaptation, and so data-efficient methods are key. Task relabelling, like in multi-task or goal-conditioned RL, makes a lot of sense in this context.

The particular proposed method seems reasonable, although I have some concerns about the detail of the exposition – I found section 4.1 fairly difficult to follow.
First, it’s not 100% clear to me how to map eq(1) to the objective written in terms of utilities because eq(1) does not define where tau_pre come from. Presumably this is just following pi_theta, and the conditioning of q(tau | psi) on psi only results in differing rewards in tau, not differing state-action sequences?
Then, most importantly for understanding this section, I don’t follow why the objective for (theta, phi) should be maximised by adjusting this q for fixed (theta, phi). The paper says this “facilitates alignment with the goals of the meta-learner” but I’m not sure what this means.
The derivation then continues in a very brusque manner. I’m not a fan of “it is easy to show”: in general if it is easy rather write it yourself (in an appendix if need be for space) or cite appropriately. Eventually we arrive at an ‘optimal’ relabeling distribution but I don’t understand in what sense it is optimal due the previous confusion.
It could be I’m missing something simple or these things are all straightforward and clear to a reader with the right context.
However, I encourage the authors to substantially clarify and elaborate this section to engage with a broad audience.

The issue that I have more intuitively with the method is that the optimal task inference should depend on the true distribution of tasks. By altering this distribution through relabeling, it seems it would change the optimal (theta, phi). Can the authors elaborate on whether or not this should be a consideration, perhaps by clarifying the exposition given in S4.1?

The implementation of the approach is quite neat. I like the use of PEARL’s particular type of value function to efficiently estimate the value of the post-adaptation policy without sampling any fresh transitions.

I also like the empirical study. The performance gains seem substantial in several tasks, and I appreciate the credible baselines which are more naive but not just vanilla PEARL without any task relabeling.
I appreciated the informative ablations.

Minor comments or questions:
 - Paragraph 2 of the intro says meta-RL is "inherently on-policy": this is incorrect.
 - Why relabel with just one task sampled from q(psi|tau)? Why not several samples, or weighted samples?
 - In Algo 1, maybe use a different letter to distinguish N in GetLogPartition and in N in ComputeUtility
 - The paper would benefit from more details on the setup with a learned reward function.

----------------------

The authors were able to clarify the points that had confused me in my initial reading. I am persuaded that the optimal meta-learned solution will not be biased by the proposed relabelling; and that the derivation is sound. Optimising the relabelling distribution for the immediate post-adaptation returns makes sense as a somewhat myopic heuristic to accelerate meta-learning.
I also appreciate the additional experiment carried out for HrwG. Further, while the connection to prior work is close in many ways, I believe the adaptation of the method for this context is sufficiently novel and effective to warrant acceptance.

**Summary Of The Paper:**

This paper studies task relabelling in hindsight to increase the efficiency of meta-reinforcement-learning.
The authors propose a strategy for calculating a distribution of tasks for which a particular batch of data would be useful for adaptation, and sample from this distribution to construct a relabelled batch which augments the training data.
The authors show empirically that this improves sample efficiency over more naive relabelling schemes, particularly for sparse reward tasks.
A series of ablations further justifies several design decisions or investigates robustness to hyperparameters.

**Summary Of The Review:**

The work is well-motivated intuitively, but the mathematical justification for the specific method is difficult to follow (so I cannot quickly verify its soundness).
The empirical study is well done overall, so I lean to accept the paper but would likely increase my score and confidence if the authors can clarify the theoretical motivation for their relabeling strategy.

---

> ### Author Response · Authors · 2021-11-22
> **Response to the Reviewer YEZ4**
>
> Thank you for your thoughtful comments and suggestions! We really appreciate your efforts and time investment, and your positive evaluation of our work. We hope to clarify any points of confusion.
>
>
> **It’s not 100% clear to me how to map eq(1) to the objective written in terms of utilities because eq(1) does not define where $\tau_\text{pre}$ come from. Presumably this is just following $\pi_\theta$, and the conditioning of $q(\tau | \psi)$ on $\psi$ only results in differing rewards in $\tau$, not differing state-action sequences:** This is correct. For a given task $\psi$, $\tau_\text{pre}$ is generated by simply following $\pi_\theta$, and the conditioning of $q(\tau | \psi)$ on $\psi$ denotes that the rewards in the collected trajectory $\tau$ are for this specific task $\psi$.
>
>
> **Then, most importantly for understanding this section, I don’t follow why the objective for (theta, phi) should be maximised by adjusting this q for fixed (theta, phi). The paper says this “facilitates alignment with the goals of the meta-learner” but I’m not sure what this means:** We answer this by building onto the response to the previous question, where we noted that the original meta-RL objective (Equation 1) could be written as OBJ := $\max_{\theta, \phi}E_{\psi \sim p(\psi)}E_{\tau \sim q(\tau | \psi)}[U_{\psi}(\tau)]$. Written this way, the distribution of pre-adaptation trajectories used for each task $q(\tau | \psi)$ is made **explicit**, and in the standard meta-RL algorithms, $q(\tau | \psi)$ is equivalent to how the reviewer described it in the previous question.
>
> Our purpose of exposing $q(\tau | \psi)$ to the optimization is to induce a more efficient distribution of pre-adaptation trajectories for a given task $\psi$ -- the specific way we achieve this is by reusing trajectories generated for other tasks, for this task $\psi$, under suitable conditions of utility. Altering this trajectory distribution does not change the theoretical optimal solution to the meta-RL problem, as we discuss in the next question. Rather, the motivation for this is to improve the sample efficiency of the meta-training phase.
>
> To optimize for $q(\tau | \psi)$, we find that a simple approach is to use the same objective OBJ (for fixed meta-parameters and augmented with entropy regularization). What we mean by “alignment with the goals of the meta-learner” is that now, both the objectives for the meta-parameters and that for $q(\tau | \psi)$ aim to maximize the post-adaptation returns for the given task distribution.
>
>
> **“I’m not a fan of “it is easy to show”: in general if it is easy rather write it yourself (in an appendix if need be for space) or cite appropriately.”:** We apologize for omitting the details here. The manuscript has been revised and the corresponding derivation has been added to Appendix A.10.
>
>
> **The issue that I have more intuitively with the method is that the optimal task inference should depend on the true distribution of tasks. By altering this distribution through relabeling, it seems it would change the optimal (theta, phi).:** Note that by relabeling data, we are not altering the true task distribution $p(\psi)$ that the meta-learner is trained on, but rather, we are sharing trajectories across the training tasks. Therefore, this does not alter the optimal (theta, phi) point under the assumption of a sufficiently exploratory agent. Consider, for simplicity, a meta-RL problem with two training tasks $T_1$ and $T_2$, where we relabel a certain amount of trajectories generated from $T_2$ with the reward function from $T_1$ and use them for meta-training on $T_1$. If we were to do no relabeling at all, in the limit of infinite exploration, we would eventually generate those $T_2$ (relabeled) trajectories while collecting data for  $T_1$ as well. In practice, we want to meta-learn policies without using a large number of samples, and relabeling helps greatly with improving the sample efficiency of meta-RL.
>
>
> **Meta-RL is not inherently on-policy:** Thank you! We have corrected this in the revised version.
>
> **Why relabel with just one task:** We relabel with just one task for simplicity and to minimize memory usage. It is certainly possible that relabeling with multiple tasks or using weighted samples could lead to improved performance, but we find that relabeling with just one task already leads to good performance.
>
> **In Algo 1, maybe use a different letter to distinguish N in GetLogPartition and in N in ComputeUtility:** Thank you for this suggestion! We have updated the manuscript with this change.
>
> **The paper would benefit from more details on the setup with a learned reward function:** Thank you for this suggestion! We have added Appendix A.7 to include these details.

---

> ### Author Response · Authors · 2021-11-30
> **Thank you for the update**
>
> We really appreciate the reviewer for their positive evaluation of our work and for taking our rebuttal into consideration in raising their rating of the paper. Thank you!

---

### Official Review · Reviewer_HrwG · 2021-11-03

**Correctness:** 4
**Technical Novelty And Significance:** 3
**Empirical Novelty And Significance:** 3
**Recommendation:** 6
**Confidence:** 5

**Main Review:**

Overall I think this paper presents an interesting idea for sharing data between tasks of a meta-RL problem. The paper is well written and the ideas are presented clearly.

Pros:

1\. I find the main insight of the paper simple and intuitive. The idea that we need to relabel not according to how much return we achieve but according to how much information we can gather for task identification gives us a clear distinction between multi-task RL and meta-RL. The derivation of relabeling according to the exponentiated post adaptation return follows naturally.

2\. The ablation study in the paper is very informative. The ablation study gives us clear comparisons, showing us which component is more important. From the ablation study, it seems that using the partition function and softmax relabeling distribution are the most important components.

3\. The paper is well written. The insights, ideas, algorithms and experiments are easy to follow.


Cons:

1\. I am somewhat skeptical about the approach of using the learned Q function to estimate return after adaptation. In the base meta-RL algorithm PEARL, the context encoder is trained to identify the task from a distribution of tasks, producing a posterior distribution of context z corresponding to that task. This means that given the relabeled trajectory, even if the context encoder predicts the context corresponding to a wrong task, as long as the produced context is within the distribution of tasks, the expected return will still be high because the policy is trained to do well also on that wrong task. Therefore it is not clear to me why using the learned Q function is a good way to estimate return on that specific task.

In order to verify this, I’d like to ask the authors to include the following experiments. First train the proposed algorithm to convergence and freeze the weights of the context encoder. Then train a N-way classifier on top of the context encoder to classify the context into one of N training tasks without using any relabeled trajectories, and then report the accuracy of the classifier. Finally relabel the trajectories according to the proposed method and report the classifier’s accuracy on top of the relabeled trajectories. If the relabeling mechanism using the learned Q function is correctly capturing the task information, we would see that the classifier’s accuracy on the relabeled trajectories is comparable to that on the true trajectories. In fact, this experiment could also lead to an even simpler relabeling strategy: directly use the true task’s probability under the classifier’s prediction as the source of relabeling signal.


2\. The empirical performance of the proposed method does not seem very strong. Only in 3 of 5 sparse reward tasks the proposed method significantly outperforms the baselines, and the proposed method does not show much improvement on dense reward tasks.


Given these limitations, I’m leaning slightly towards not accepting the paper. I’d highly encourage the authors to conduct the experiment I suggested in order to verify that the proposed method is indeed capturing the task information correctly.


## Update After Author Response
The authors conducted additional classifier experiments I requested and the results suggest that using learned Q function to estimate returns is highly informative about the task. Therefore my main concern about the proposed method has been addressed, and I'm now leaning towards accepting the paper.



**Summary Of The Paper:**

This paper proposes a way to share data across different tasks in meta-reinforcement learning (meta-RL), where the data from one task is reused in another task by relabeling the rewards. Based on the HIPI method ([1]), the authors construct a relabeling distribution to relabel the pre-adaptation trajectories from one task to be used for another task. The relabeling probability of a trajectory is chosen to be proportional to the exponentiated utility function, which is defined as the expected return after the agent uses that trajectory to adapt. In practice, the post-adaptation return is approximated using the learned Q function. The authors apply this relabeling distribution to PEARL, an existing off-policy actor-critic style meta-learning algorithm.

The authors conduct experiments on simulated robotics experiments. The results suggest that the proposed method outperforms prior methods on sparse reward tasks, while performing roughly the same on dense reward tasks.


References

[1] Eysenbach, Benjamin, et al. "Rewriting history with inverse rl: Hindsight inference for policy improvement." arXiv preprint arXiv:2002.11089 (2020).


**Summary Of The Review:**

The paper presents an interesting idea about reusing data across tasks in meta-RL. The idea is very intuitive and the paper is well written. However, I’m not sure about whether the approach used to implement the idea of the paper really does what the authors claim it does. Therefore I’d like to see more evidence before I can recommend accepting the paper.

---

> ### Author Response · Authors · 2021-11-22
> **Response to the Reviewer HrwG**
>
> Thank you for your thoughtful comments and suggestions! We hope to address the concerns raised by the reviewer:
>
> **Skeptical about the approach of using the learned Q function to estimate return after adaptation. Would like to see experiment to verify:** Thank you for the excellent suggestion! We have added the proposed experiment to Appendix A.9. We note that the accuracy of the N-way classifier is actually **higher** for the relabeled trajectories than for the non-relabeled trajectories. We believe this is potentially because several non-relabeled trajectories are uninformative for task-identification for the task they were originally generated for, which is precisely the problem that the HFR relabeling attempts to address. We hope this new experiment verifies that the relabeling mechanism using the context encoder and the learned Q function is correctly capturing the task information. The results are summarized below for convenience; the experimental setup details are included in Appendix A.9.
>
>
>
> | Environment        | Accuracy on Relabeled Trajectories     | Accuracy on Non-Relabled Trajectories |
> |:-------------------:|:---------:|:---------------:|
> | Four-Corners           | 89.65% | 76.98%       |
> | Cheetah-Vel-Sparse    | 74.33% | 57.58%      |
>
>
> Very much connected to this -- we would also like to refer the reviewer to Figure 11 and Table 3 in Appendix A.6. Figure 11 contains a set of example trajectories on our didactic Four-Corners environment, while the HFR column of Table 3 shows the expected post-adaptation returns (as measured by the learned Q function) after using the relabeled example trajectories for adaptation on each task. For all 4 trajectories, the post-adaptation Q-value is highest for the correct task, and for 3/4 trajectories this value for the correct task is _considerably_ higher than that for the incorrect tasks.
>
> **Simpler relabeling strategy using task-classifier predictions:** We agree with the reviewer that this is an intriguing direction for future work. Directly considering how useful a trajectory is for task identification makes a lot of sense, and we did experiment along these lines in our paper by using the Bellman error under various tasks as the source of relabeling signal (please see Appendix A.1). We find that this Bellman variant performs worse than HFR, likely because HFR directly considers the post-adaptation return, which is the metric of interest, while the Bellman error is simply an indirect measure of post-adaptation return.
>
> **Empirical performance does not seem very strong:** We would like to highlight that HFR comfortably outperforms baselines on 4/5 sparse-reward tasks (Ant-Goal, Sawyer-Push, Cheetah-Vel-Sparse, Visual-Reacher) and 1 dense reward task (Cheetah-Highdim), although the performance benefit is the most pronounced for Sawyer-Push. The improvements are probably not communicated sufficiently well due to the figure sizes/scales. To remedy that, we have included below the final numerical returns/final success rate of HFR and the **best baseline** on each task and reported the percentage improvement over the best baseline. We believe these improvements are significant, especially given that HFR is a lightweight addition that doesn’t modify the underlying PEARL algorithm.
>
> | Environment        | HFR     | Best Baseline | Improvement over best baseline |
> |:--------------------:|:---------:|:---------------:|:---------------:|
> | Ant-Goal           | 400.62  | 331.76        | 21%      |
> | Cheetah-Highdim    | 3551.50 | 2987.73       | 19%      |
> | Cheetah-Vel-Sparse | 254.19  | 212.35        | 20%      |
> | Sawyer-Push        | 0.82    | 0.45          | 82%      |
> | Visual-Reacher     | 10.89   | 8.81          | 23%      |
>
> **Proposed method does not show much improvement on dense reward tasks:** As discussed in Subsection 5.1, we believe that this is because exploration is not as critical for most of these tasks, with the exception of Cheetah-Highdim, where HFR does yield substantial performance improvement (19% over the best baseline). This is supported by the fact that in dense-reward environments where HFR does not provide big gains, the other relabeling methods achieve similar performance to HFR, as does PEARL with no relabeling. Nonetheless, HFR does yield significant improvements in sparse reward meta-RL tasks. Given the difficulty of designing appropriate dense reward functions compared to the ease of specifying a simple sparse reward function, we believe the improvements provided by HFR in sparse reward settings have notable value.

---

> > ### Comment · Reviewer_HrwG · 2021-11-22
> > **Re: Author Response**
> >
> > First of all I'd like to thank the authors for the addition of the experiment I requested. The results of that experiment is very convincing and thus verifies the approach of using learning Q functions to estimate the post adaptation return. This addresses my main concerns and now my skepticism about the approach is cleared. I will increase my rating of the paper.

---

> > > ### Author Response · Authors · 2021-11-23
> > > **Thank you for for your response**
> > >
> > > We really appreciate the reviewer for taking our rebuttal into account and raising their rating of the paper. Thank you!

---

### Official Review · Reviewer_P7KG · 2021-11-06

**Correctness:** 4
**Technical Novelty And Significance:** 2
**Empirical Novelty And Significance:** 3
**Recommendation:** 6
**Confidence:** 4

**Main Review:**

Strengths

1. Usefulness of relabelling

The problem considered is an important one, since even though hindsight relabelling is standard in multi-task RL, and has been shown to enable learning on sparse reward environments (which are otherwise very difficult to solve), this approach hasn’t been applied to the meta-RL setting yet. This is despite the fact that meta-RL also considers a multi-task distribution, and can benefit from explicitly using data for a different task and relabelling it under the corresponding reward function. The mathematical formulation of the approach closely follows HIPI[2], with the difference that post-adaptation trajectory return is considered instead of current trajectory return, to be aligned with the meta-learning objective. The authors show experimentally that current meta-RL approaches do not begin to make progress on sparse-reward tasks, showing the importance and effectiveness of relabelling.

2. Extent of Evaluation and Analysis

The paper includes evaluation on 5 different sparse reward environments, 3 dense reward environments which show that the relabelling scheme offers benefits over current meta-RL approaches mainly in the sparse reward setting. The authors also include ablations/analysis of specific components, such as using a learned reward instead of the true reward, using hardmax instead of softmax for sampling the relabelling task etc. The paper is well written, the presentation is clear and well-motivated.

Weaknesses

1. Small performance gap with HIPI, Simplistic Environments

Out of the 8 experimental domains chosen, the performance of the proposed approach (HFR) is significantly better than HIPI on only two domains (ant-goal and sawyer-push). This indicates that most of the benefit is coming from the relabelling scheme for most environments, either because the adaptation procedure doesn't actually lead to better performance, or because the environments are too simple to require a lot of adaptation to held-out tasks. Given that performance is much better than HIPI on the hardest environments (ant-goal and sawyer-push), I am inclined to think the issue is the latter instead of the former, which can be addressed by evaluating on harder environments.

These could include other single-family robotic tasks from meta-world (eg: sawyer-door-open, sawyer-box-close etc). Even better would be meta-training across task families, using MetaWorld ML-10 or ML-45. This would test adaptation to tasks that are semantically different, and would make the paper a lot more compelling.


**Summary Of The Paper:**

The paper proposes an approach for using data relabelling for meta-RL for better sample efficiency and to enable training on sparse reward environments. Specifically the proposed method combines Pearl[1] with a modified version of HIPI[2], where the trajectories chosen for relabelling are effective for adaptation, and not necessarily high in reward themselves.

[1] : Efficient Off-Policy Meta-RL vis Probabilistic Context Variables (Rakelly et al.)
[2] : Rewriting History with Inverse RL (Esyenbach et al.)






**Summary Of The Review:**

The approach introduces relabelling (which has already shown to be important in multi-task RL) in meta-RL, and shows superior performance on sparse reward environments. The paper would be more compelling if it included evaluations on more challenging environments, to establish the importance of the adaptation component.

---

> ### Author Response · Authors · 2021-11-22
> **Response to the Reviewer P7KG**
>
> Thank you for your thoughtful comments and suggestions! We really appreciate your positive evaluation of our work. We address the following concerns raised by the reviewer:
>
> **Small performance gap with HIPI (HFR significantly better only on ant-goal and sawyer-push):** We would like to highlight that HFR comfortably outperforms baselines on 4/5 sparse-reward tasks (Ant-Goal, Sawyer-Push, Cheetah-Vel-Sparse, Visual-Reacher) and 1 dense-reward task (Cheetah-Highdim), although the performance benefit is the most pronounced for Sawyer-Push. The improvements are probably not communicated sufficiently well due to the figure sizes/scales. To remedy that, we have included below the final numerical returns/final success rate of HFR and the **best baseline** on each task and reported the percentage improvement over the best baseline. We believe these improvements are significant, especially given that HFR is a lightweight addition that doesn’t modify the underlying PEARL algorithm.
>
> | Environment        | HFR     | Best Baseline | Improvement over best baseline |
> |:--------------------:|:---------:|:---------------:|:---------------:|
> | Ant-Goal           | 400.62  | 331.76        | 21%      |
> | Cheetah-Highdim    | 3551.50 | 2987.73       | 19%      |
> | Cheetah-Vel-Sparse | 254.19  | 212.35        | 20%      |
> | Sawyer-Push        | 0.82    | 0.45          | 82%      |
> | Visual-Reacher     | 10.89   | 8.81          | 23%      |
>
>
>
> **Most of the benefit is coming from the relabeling scheme for most environments rather than the adaptation procedure:** We would like to clarify that our contribution is strictly a new relabeling scheme for meta-RL. We neither introduce a new adaptation procedure nor modify the adaptation procedure of the base meta-RL algorithm (Please refer to Figure 4). The adaptation procedure is the same across HFR and all baselines (HIPI, Random, None) and consists of a learned context encoder in our experiments, although it is true that different relabeling schemes will cause the learned context encoder to be different. Since the only difference between HFR and the baseline points of comparison is the relabeling scheme, it is expected that the performance difference can be attributed solely to the relabeling scheme.
>
> **Simplistic environments:** To create the task distributions for evaluating HFR, we looked at the recent meta-RL literature [1,2,3,4,5,6,7] and assembled the various environments from these papers. The tasks we evaluate on encompass a variety of robots (Sawyer robot, Ant, Cheetah) and a variety of different behaviors (pushing, goal-reaching, locomotion at various speeds, locomotion to various targets).
>
> We agree with the reviewer that Meta-world is a useful testbed for meta-RL tasks. However, Meta-world considers a task distribution with varying transition dynamics, whereas we consider trajectory relabeling in the setting where the reward function differs amongst the meta-training tasks but the transition dynamics are the same. We would like to point the reviewer to Appendix A.2 in our paper, where we show results on environments where the tasks differ in both the dynamics and the reward functions (Ant-goal-rand-params, Cheetah-vel-sparse-rand-params). Naturally, these are harder scenarios than those considered in Figure 6 and Figure 7. Surprisingly, we see some performance gains over the baselines without modifying HFR to adapt to this setting. Nonetheless, designing a more principled approach and extending HFR to be compatible with the Meta-world distributions is important future work.
>
>
>
>
> [1] Meta-Reinforcement Learning Robust to Distributional Shift via Model Identification and Experience Relabeling (Mendonca et al.)
>
> [2] Offline Meta Learning of Exploration (Dorfman et al.)
>
> [3] Efficient Off-Policy Meta-Reinforcement Learning via Probabilistic Context
> Variables (Rakelly et al.)
>
> [4] Model-Agnostic Meta-Learning for Fast Adaptation of Deep Networks (Finn et al.)
>
> [5] RL2: Fast Reinforcement Learning via Slow Reinforcement Learning (Duan et al.)
>
> [6] VariBAD: A Very Good Method for Bayes-Adaptive Deep RL via Meta-Learning
> (Zintgraf et al.)
>
> [7] Meta-Q-Learning (Fakoor et al.)

---

### Author Response · Authors · 2021-11-22
**Revised Paper**

We would like to thank the reviewers for spending the time to review the paper and provide quality feedback with constructive suggestions for improvement. We address each reviewer's comments separately below, and summarize the major changes made in the revision here:

1. Added Appendix A.7 to include details about how we learn a reward function in our ablation study (Section 5.2).

2. Added Appendix A.8 with algorithm bar for the meta-test phase.

3. Added Appendix A.9 to include the experiment suggested by Reviewer HrwG to verify that using the learned Q function is correctly capturing task information.

4. Added Appendix A.10 to show equivalence between Equation 5 and the reverse KL-divergence minimization objective.

5. Typos and other clarifications/re-wordings have been included in the revised manuscript, as per the suggestions of all the reviewers.


All additions are highlighted using blue-colored text in the revision.

---

### Author Response · Authors · 2021-11-29
**Any further follow-ups?**

We want to thank all the reviewers once again for their time and effort in helping us improve our paper. Given that the final stage of discussion ends on 11/29, we’d like to ask if any of the reviewers have any remaining questions?

---

### Decision · Program_Chairs · 2022-01-20

**Decision:**

Accept (Poster)

**Comment:**

This paper proposes Hindsight Foresight Relabeling (HFR), an approach for reward relabeling for meta RL. The main contribution is a measure of how useful a given trajectory is for the purpose of meta-task identification as well as the derivation of a task relabeling distribution based on this measure.

Reviewers agreed that the paper tackles an interesting problem and found the main insight to be simple and intuitive. While the initial reviews raised some concerns regarding novelty, the performance gap, and using the learned Q-function to estimate post-adaptation returns the rebuttal did a good job of addressing these concerns. Overall, the paper proposes a non-trivial extension of hindsight relabeling to meta RL and while the results could be stronger I think the paper provides useful ideas and insights so I recommend acceptance as a poster.